# FAIRNESS IMPROVES LEARNING FROM NOISILY LABELED LONG-TAILED DATA

## ABSTRACT

Both long-tailed and noisily labeled data frequently appear in real-world applications and impose significant challenges for learning. Most prior works treat either problem in an isolated way and do not explicitly consider the coupling effects of the two. Our empirical observation reveals that such solutions fail to consistently improve the learning when the dataset is long-tailed with label noise. Moreover, with the presence of label noise, existing methods do not observe universal improvements across different sub-populations; in other words, some sub-populations enjoyed the benefits of improved accuracy at the cost of hurting others. Based on these observations, we introduce the **F**airness **R**egularizer (**FR**), inspired by regularizing the performance gap between any two sub-populations. We show that the introduced fairness regularizer improves the performances of sub-populations on the tail and the overall learning performance. Extensive experiments demonstrate the effectiveness of the proposed solution when complemented with certain existing popular robust or class-balanced methods.

## 1 INTRODUCTION

Biased and noisy training datasets are prevalent and impose challenges for learning (Salakhutdinov et al., 2011; Zhu et al., 2014; Liu, 2021). The biases and noise can happen both at the sampling and label collection stages: A dataset often contains numerous sub-populations and the size of these sub-populations tends to be long-tailed distributed (Salakhutdinov et al., 2011; Zhu et al., 2014), where the tail sub-populations have an exponentially scaled probability of being under-sampled. Meanwhile, a dataset tends to suffer from noisy labels if collected from unverified sources (Wei et al., 2022c). Most prior works treat either population bias or label noise in an isolated way and do not explicitly consider the coupling effects of the two. In particular, existing works on learning with noisy labels mainly focus on a homogeneous treatment of the entire population, and the underlying clean data is often balanced (Natarajan et al., 2013; Liu & Tao, 2015; Patrini et al., 2017; Liu & Guo, 2020; Cheng et al., 2021).

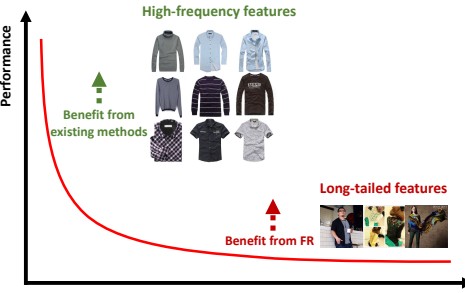

Figure 1: Overview of our work: Different robust solutions incur varied impacts on noisily labeled long-tailed distributed sub-populations. We show adding **F**airness **R**egularizer (**FR**) between head and tail populations encourages the classifier to achieve relatively fair performances by reducing performance gaps among sub-populations, and improves the overall learning performance.

The main inquiry of our paper is to understand and mitigate the possible heterogeneous effects of label noise when considering the imbalanced distribution of sub-populations. We start by presenting strong evidence of disparate impacts of sub-populations with a synthetic long-tailed noisy CIFAR-100 dataset (Krizhevsky et al., 2009) when using existing learning with noisy labels methods. Figure 2 illustrates the per-population (100 sub-populations in all, where we consider the class information as a natural separation of sub-populations) performance comparisons between applying the traditional Cross-Entropy (CE) loss and the recently proposed robust treatment to either noisy (i.e., Label Smoothing (LS) (Lukasik et al., 2020) and PeerLoss (PL) (Liu & Guo, 2020)) or long-tailed data (Focal (Lin et al., 2017), Logit-adjustment (Menon et al., 2021)). There are three main takeaways:

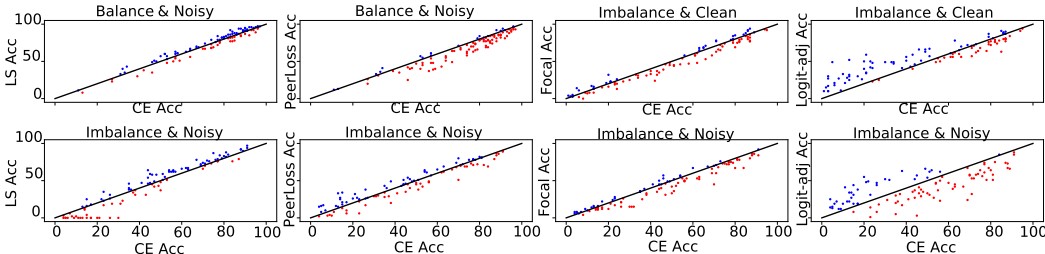

Figure 2: How each method improves per sub-population test accuracy w.r.t. CE loss on CIFAR-100 dataset. All methods are trained on 20 coarse classes in CIFAR-100. Each coarse class includes 5 different sub-populations (fine classes in CIFAR-100). For each sub-figure, $x$-axis indicates the CE accuracy. $y$-axis denotes the performance of robust/long-tail approaches. Each dot denotes the test accuracy pair $(\text{Acc}_{CE}, \text{Acc}_{Method})$ for each sub-population. The line $y = x$ means that CE performs the same as the robust treatment on a particular sub-population. The blue (red) dot above (below) the line shows the robust treatment has positive (negative) effect on this sub-population compared with CE. In sub-titles, "Balance" denotes the balanced training data (w.r.t. clean labels); "Imbalance" means the training dataset follows a long-tailed distribution where the ratio between max and min number of samples in the sub-populations is 100; "Clean": the labels of training samples are clean; "Noisy": 25.6% training samples are wrongly labeled. The test dataset is clean and balanced.

*First*, the same robust treatment may have disparate impacts on different sub-populations, e.g., different sub-populations are improved differently by losses such as the Logit-adj loss (Menon et al., 2021). *Second*, different robust treatments have disparate impacts on the same part of data, e.g., LS (Lukasik et al., 2020) performs badly (almost 0 accuracies) on sub-populations with low CE accuracy (<50) and improves the others, while PL (Liu & Guo, 2020) has a reversed effect that the high CE accuracy part (>50) is likely to be degraded. *Third*, the prior works fail to address the coupling effects of population imbalance and noisy labels.

The above observations motivate us to explore how sub-population data should be treated when learning from noisily labeled long-tailed data. This work formally investigates the influence of sub-populations when learning with long-tailed and noisily labeled data. The analysis inspires us to define a fairness regularizer for this learning task. Figure 1 overviews our work. Our contributions are primarily two-fold. We quantify the influence of sub-populations using a number of metrics and discover disparate impacts of long-tailed sub-populations when label noise presents (Section 3). Following the above observation, we propose the **F**airness **R**egularizer (**FR**), which encourages the learned classifier to reduce the performance gap between the head and tail sub-populations. As a result, our approach not only improves the performances of tail populations but also improves overall learning performance. Extensive experiments on the CIFAR and Clothing1M datasets demonstrate the effectiveness of **FR** when complemented with certain robust or long-tailed solutions (Section 5).

Contrary to most existing fairness-accuracy trade-offs observed in the literature (Hardt et al., 2016; Menon & Williamson, 2018; Martinez et al., 2019; Zhao & Gordon, 2019; Ustun et al., 2019; Islam et al., 2021), we show that adding this fairness regularizer alleviates disparate impacts across populations of different sizes and improves the learning from noisily labeled long-tailed data.

## 1.1 RELATED WORKS

**Learning with Noisy Labels** Obtaining perfect annotations in supervised learning is a challenging task (Xiao et al., 2015; Luo et al., 2020; Wei et al., 2022c;d). Due to the restrictions of human recognition, noisy annotations impose challenges to performing robust training. A line of popular approaches of learning with label noise firstly estimates the noise transition matrix, and then proceeds to use this knowledge to perform loss or sample correction (Jiang et al., 2022; Natarajan et al., 2013; Liu & Tao, 2015; Patrini et al., 2017; Zhu et al., 2021b; Li et al., 2022), i.e., the surrogate loss uses the transition matrix to define unbiased estimates of the true losses (Scott et al., 2013; Natarajan et al., 2013; Scott, 2015; Van Rooyen et al., 2015; Menon et al., 2015). Noting that the estimation of the noise transition matrix is non-trivial (Zhu et al., 2021b; 2022b), another line of works aims to propose training methods without requiring knowing the noise rates, e.g., using robust loss functions (Kim et al., 2019; Liu & Guo, 2020; Wei & Liu, 2020; Wei et al., 2022b) training deep neural nets directly without the knowledge of noise rates (Han et al., 2018; Wei et al., 2020; 2022a; Qin et al., 2022),

making use of the early stopping strategy (Liu et al., 2020; Xia et al., 2021a; Liu et al., 2022a;b; Huang et al., 2022), or designing a pipeline which dynamically selects/corrects and trains on "clean" samples with small loss (Cheng et al., 2021; Xia et al., 2021b; 2022b; Jiang et al., 2022; Zhang et al., 2022a). Recent works also explored the possibility of using open-set data to improve the closed-set robustness (Wei et al., 2021a; Xia et al., 2022a).

**Learning with Long-Tailed Data**   The most relevant mainstream solution of learning with long-tailed clean data is the logit/loss adjustment approaches, which modify the loss values during the training procedure, for example, adjust the loss values w.r.t. the label frequency (Ren et al., 2020), sample influence (Park et al., 2021), or the distribution alignment between model prediction and a set of the balanced validation set (Wei et al., 2023), among many other solutions. More recently, based on the label frequencies, the logit adjustments over classic approaches (Menon et al., 2021) are proposed, either through a post-hoc modification w.r.t. a trained model or enforcement in the loss during training. Open-set data may also be used to improve complement long-tailed data (Wei et al., 2021a). Please refer to a comprehensive survey (Zhang et al., 2023) for more details.

Existing robust approaches targeted mainly the class or sub-population level balanced training data. More recently, the literature observed several approaches to address the issue of label-noise in the long-tailed tasks, through decoupled treatments for head classes and tail ones, i.e., detecting noisy labels and performing robust solutions to the head class, meanwhile adopting a self/semi-supervised learning manner to deal with the tail classes (Zhong et al., 2019; Wei et al., 2021c; Karthik et al., 2021). Beyond classes, it has been demonstrated that sub-populations with different noise rates cause disparate impacts (Liu, 2021; Zhu et al., 2022a) and need decoupled treatments (Zhu et al., 2021a; Wang et al., 2021), which is more crucial for long-tailed sub-populations.

## 2 PRELIMINARY

### 2.1 SUB-POPULATIONS OF FEATURES

In a $K$-class classification task, denote a set of data samples with clean labels as $S := \{(x_i, y_i)\}_{i=1}^n$, given by random variables $(X, Y)$, which is assumed to be drawn from $\mathcal{D}$. In this work, we are interested in how sub-populations intervene with learning. Formally, we denote $G \in \{1, 2, ..., N\}$ as the random variable for the index of sub-population, and each sample $(x_i, y_i)$ is further associated with a $g_i$. The set of sub-population $k$ could then be denote as $\mathcal{G}_k := \{i : g_i = k\}$. We consider a long-tail scenario where the head population and the tail population differ significantly in their sizes, i.e., $\max_k |\mathcal{G}_k| \gg \min_{k'} |\mathcal{G}_{k'}|$.

Consider Figure 3 for an example of sub-population separations using the CIFAR-100 dataset (Krizhevsky et al., 2009): images are grouped into 20 coarse classes, and each coarse class could be further categorized into 5 fine classes. For example, the coarse class "aquatic mammals" was further split into "beaver", "dolphin", "otter", "seal", and "whale". From Figure 3, we observe a strong imbalanced distribution of different sub-populations and a long-tailed pattern. In Section 5.1, we provide more details on long-tail data generation models for our synthetic experiments.

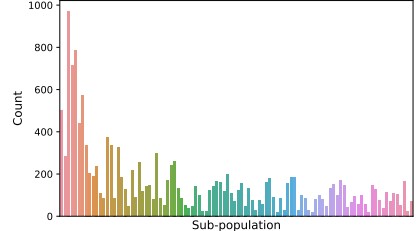

Figure 3: Count plot of a synthetic long-tailed CIFAR-100 train dataset: $x$-axis denotes the sub-population index; $y$-axis indicates the number of samples in each sub-population.

### 2.2 OUR TASK

In practice, obtaining "clean" labels from human annotators is both time-consuming and expensive. The obtained human-annotated labels usually consist of certain noisy labels (Xiao et al., 2015; Lee et al., 2018; Jiang et al., 2020; Wei et al., 2022c). The flipping from clean labels to noisy labels is usually described by the noise transition matrix $T(X)$, with its element denoted by $T_{ij}(X) = \mathbb{P}(\widetilde{Y} = j|Y = i, X)$. We denote the obtained noisy training dataset as $\widetilde{S} := \{(x_i, \tilde{y}_i)\}_{i=1}^n$, given by random variables $(X, \widetilde{Y})$, which is assumed to be drawn from $\widetilde{\mathcal{D}}$.

Though we only have access to noisily labeled long-tailed data $\widetilde{S}$, our goal remains to obtain the optimal classifier with respect to a clean and balanced distribution $\mathcal{D}$: $\min_{f \in \mathcal{F}} \mathbb{E}_{(X,Y) \sim \mathcal{D}} [\ell(f(X), Y)]$, where $f$ is the classifier chosen from the hypothesis space $\mathcal{F}$, and $\ell(\cdot)$ is a calibrated loss function (e.g., CE) Furthermore, we do not assume the knowledge of the sub-population information during training. We are interested in how sub-populations intervene with the learning performance and how we could improve by treating the sub-populations with special care.

## 3 DISPARATE INFLUENCES OF SUB-POPULATIONS: AN EMPIRICAL STUDY

In this section, we empirically illustrate the disparate influence of sub-populations when learning with noisily labeled data. Inspired by the literature on using the influence function to capture the impact of training samples, we define influence metrics at the sub-population level and perform a multi-faceted evaluation of how imbalanced sub-populations affect the learning performance. We take the long-tail populations for illustration and defer the results of head populations to Appendix C.4.

**Influences:** In the literature of explainable deep learning, the notions of influence can be different, e.g., the influences of features on an individual sample prediction (Ribeiro et al., 2016; Sundararajan et al., 2017; Lundberg & Lee, 2017; Feldman & Zhang, 2020), the influences of features on the loss/accuracy of the model (Owen & Prieur, 2017; Owen, 2014), the influences of training samples on the loss/accuracy of the model (Jia et al., 2019). In this section, we focus on the influence of a sub-population on both the sub-population level and the individual sample level.

We now empirically demonstrate the role of sub-populations when measuring the test accuracy, and the prediction of model confidence on test samples. For the synthetic long-tailed noisy training dataset, we first flip clean labels of the class-balanced CIFAR-10 dataset to any other classes, and there exist 20% wrong labels in all. We then adopt the class-imbalanced (Cui et al., 2019) CIFAR-10 dataset to select a long-tailed distributed amount of samples for each class (by referring to clean labels). As for the separation of sub-populations, we adopt the $k$-means clustering to categorize the extracted features of each feature given by the Image-Net pre-trained model. Since sub-population information sometimes may not be available for training use, understanding the influences of such division of sub-populations is beneficial. More details can be found at 5.1.

We explore the influences of tail sub-populations on performances of cross-entropy (ce) loss, the forward loss correction (fw) (Patrini et al., 2017), label smoothing (ls) (Lukasik et al., 2020), and the peer loss (pl) (Liu & Guo, 2020). There are 17 sub-populations (train) with less than 50 instances considered as the tail section. We illustrate observations on several randomly selected tail sub-populations. Results of more sub-populations are deferred to Appendix C.1.

### 3.1 INFLUENCES ON SUB-POPULATION LEVEL (TEST ACCURACY)

We start with the influence of sub-populations in the test set. We adopt the (population-level) test accuracy changes when removing all samples in the sub-population $\mathcal{G}_i$ during the training procedure to capture the influences of a sub-population on each sub-population at the test set:

$$\text{Acc}_{\text{p}}(\mathcal{A}, \widetilde{S}, i, j) = \mathbb{P}_{\substack{f \leftarrow \mathcal{A}(\widetilde{S}) \\ (X', Y', G=j)}} (f(X') = Y') - \mathbb{P}_{\substack{f \leftarrow \mathcal{A}(\widetilde{S}^{\backslash i}) \\ (X', Y', G=j)}} (f(X') = Y'),$$

where in the above two quantities, $f \leftarrow \mathcal{A}(\widetilde{S})$ indicates that the classifier $f$ is trained from the whole noisy training dataset $\widetilde{S}$ via Algorithm $\mathcal{A}$, $f \leftarrow \mathcal{A}(\widetilde{S}^{\backslash i})$ means $f$ is trained on $\widetilde{S}$ without samples in the sub-population $\mathcal{G}_i$. $(X', Y', G=j)$ denotes the test data distribution given that the samples are from the $j$-th sub-population.

In Figure 4, the $x$-axis denotes the loss function for training, and the $y$-axis visualized the distribution of $\{\text{Acc}_{\text{p}}(\mathcal{A}, S, i, j)\}_{j \in [100]}$ (left) and $\{\text{Acc}_{\text{p}}(\mathcal{A}, \widetilde{S}, i, j)\}_{j \in [100]}$ (right) for several randomly selected long-tail sub-populations ($i = 52, 70, 91$, results of more populations can be found in Appendix C.1) under each robust method, where "$S$" refers to the clean training samples and "$\widetilde{S}$" denotes the noisy version. The blue zone shows the 25-th percentile ($Q_1$) and 75-th percentile ($Q_3$) accuracy changes, and the orange line indicates the median value. Accuracy changes that are drawn as circles are viewed as outliers. Note that all sub-figures (distributions) have the same amount of samples, it is clear to observe the left three figures have lower variance than the right ones, indicating that:

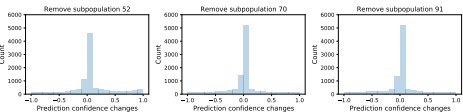 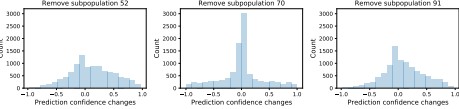

Figure 4: Box plot of the population-level test accuracy changes when removing all samples of a selected long-tailed sub-population during the training w.r.t. 4 methods. (Left: trained on clean labels; Right: trained on noisy labels.)

Figure 5: Distribution plot w.r.t. the changes of model confidence on the test data samples using CE loss (Left: trained on clean labels; Right: trained on noisy labels). See Appendix C.1 for more details.

> *Observation* 3.1. The tail sub-populations in the noisy training tend to have a more significant influence on the test accuracy than that in clean training.

## 3.2 INFLUENCES ON SAMPLE LEVEL (PREDICTION CONFIDENCE)

Note that grouping testing samples into classes/sub-populations for analysis may ignore some individual behavior changes, we next consider the influence of sub-populations on the individual test samples. Instead of insisting on the accuracy measure, we adopt the model prediction confidence as a proxy, to see how each test sample was influenced. And we introduce $\text{Infl}(\mathcal{A}, \widetilde{S}, i, j)$ to quantify the influence of a sub-population on a specific test sample:

$$\text{Infl}(\mathcal{A}, \widetilde{S}, i, j) = \mathbb{P}_{f \leftarrow \mathcal{A}(\widetilde{S})}(f(x'_j) = y'_j) - \mathbb{P}_{f \leftarrow \mathcal{A}(\widetilde{S} \setminus i)}(f(x'_j) = y'_j).$$

As shown in Figure 5, we visualize $\text{Infl}(\mathcal{A}, S, i, j)$ (left) and $\text{Infl}(\mathcal{A}, \widetilde{S}, i, j)$ (right), where $j \in [10000]$ means 10K test samples. For example, $\text{Infl}(\mathcal{A}, \widetilde{S}, i, j) = -1$ means the model prediction confidence on test sample $x'_j$ changed from 0 to 1. With the presence of label noise, we observe:

> *Observation* 3.2. Compared with clean training, removing certain tail sub-populations in the noisy training leads to significant changes/influences on the model prediction confidence of more test samples.

Concluding this section, we have shown that given certain robust methods, significant disparate impacts on sub-populations are observed, when learning from long-tailed data with noisy labels. Such impacts also differ when complemented with different robust solutions, i.e., robust loss functions implicitly incur disparate impacts on the populations/samples. Recall in Figure 2, we revealed that existing robust treatments may result in unfair performances among sub-populations, when learning from noisily labeled long-tailed data. All these observations motivate us to explore ways that will reduce the gaps between the head and the tail populations.

## 4 FAIRNESS REGULARIZER (**FR**)

In this section, we propose to assign fairness constraints to the learning objective function. Leveraged into its Lagrangian form, such fairness constraints could be viewed as fairness regularizers that explicitly encourage the classifier to achieve fair performances among sub-populations. We name our solution the **F**airness **R**egularizer (**FR**), which encourages the learned classifier to achieve fair performance across sub-populations.

### 4.1 FAIRNESS CONSTRAINTS

Note that when learning with robust methods, the classifier tends to result in fitting on certain sub-populations more easily. We propose to constrain the classifier's performance on sub-populations:

$$\min_{f:\text{domain}(X) \to [K]} \mathbb{E}_{(X,\widetilde{Y}) \sim \widetilde{\mathcal{D}}}[\ell(f(X), \widetilde{Y})], \qquad \text{s.t. Constraint w.r.t. } \mathbb{P}(f(X) = \widetilde{Y} \mid G = i), \quad (1)$$

where $\ell$ is a generic loss function that could be any robust losses and the ultimate goal of the classifier $f$ is categorizing the feature $X$ into a specific class within $[K]$. Since we do not wish certain sub-populations to fall much behind others, i.e., in terms of accuracy, we constrain the performance gap between any two sub-populations by adopting the following constraint for Eqn. (1), specifically, for any sub-population $i \in [N]$, we require its performance to have a bounded distance from the average performance. Define $\mathsf{Dist}_i := \left| \mathbb{P}(f(X) = \widetilde{Y} \mid G = i) - \mathbb{P}(f(X) = \widetilde{Y}) \right|$ as the distance (absolute performance gap), then the optimization problem is formulated as:

$$\min_{f:X \to [K]} \mathbb{E}_{(X,\widetilde{Y}) \sim \widetilde{\mathcal{D}}}[\ell(f(X), \widetilde{Y})], \quad \text{s.t. } \mathsf{Dist}_i \leq \delta, \forall i \in [N], \tag{2}$$

where $\delta \geq 0$ is a constant. Setting $\delta = 0$ implies that the classifier should achieve fair performances among all sub-populations, in order to satisfy the constraints.

## 4.2 Using Fairness Constraints as a Regularizer

In practice, forcing sub-populations to achieve absolutely fair or equalized performances (i.e., accuracy) may produce side effects. For example, one trivial solution to achieve $\delta = 0$ is simply reducing the performance of all the other sub-populations to be aligned with the worst sub-population, leading to poor overall performance. Even though we can fine-tune $\delta$ to set an appropriate tolerance of the gap, the sub-population with the worst performance may still violate the constraint. Noting our goal is to improve the overall performance on clean and balanced test data, it is arguably a better strategy to not over-addressing the worst sub-population.

To balance the trade-off between mitigating the disparate impacts among sub-populations and the possible negative effect due to constraining, rather than strictly solving the constrained optimization problem in Eqn. (2), we use the constraint as a regularizer by converting it to its Lagrangian form:

$$\min_{f:X \to [K]} \mathcal{L}_\lambda(f) := \mathbb{E}_{(X,\widetilde{Y}) \sim \widetilde{\mathcal{D}}}[\ell(f(X), \widetilde{Y})] + \underbrace{\sum_{i=1}^N \lambda_i \mathsf{Dist}_i}_{\to \mathbf{FR}},$$

where $\lambda_i \geq 0$. Different from the traditional dual ascent of Lagrange multipliers (Boyd et al., 2011), we fix $\lambda_i$ during our training. Intuitively, applying dual ascent is likely to result in a large $\lambda_i$ on the worst sub-population, inducing possible negative effects as we discussed above. Therefore, in such a minimization task, the accuracy/performance gaps between sub-populations are encouraged to be small and do not have to be exactly lower than any threshold. To clarify, we do not require strict fair performance among sub-populations, instead, we wish to improve the worst group performance at the minimum cost of the better group. Hence, we did explore the usage of other fairness constraints since all these definitions/constraints will serve with the same purpose – avoiding the performance gap among sub-populations from being overly large, in the setting of noisy labeled long-tailed data.

**Implementation** Denote by $\boldsymbol{f}_x[\tilde{y}]$ the model's prediction probability on the noisy label $\tilde{y}$ given input $x$. Noting the probability in $\mathsf{Dist}_i$ is non-differentiable w.r.t $f$, we apply the following empirical relaxation (Wang et al., 2022):

$$\mathsf{Dist}_i := \left| \frac{\sum_{k=1}^N \boldsymbol{f}_{x_k}[\tilde{y}_k] \cdot \mathbb{1}(g_k = i)}{\sum_{k=1}^N \mathbb{1}(g_k = i)} - \frac{\sum_{k=1}^N \boldsymbol{f}_{x_k}[\tilde{y}_k]}{N} \right|, \tag{3}$$

where $\mathbb{1}(g_k = i) = 1$ when $g_k = i$ and 0 otherwise. For simplicity, we set all $\lambda_i$ to a constant $\lambda$.

To demonstrate why **FR** helps with improving the learning from noisily labeled long-tailed data, we will provide extensive experiment studies in the next section. We also adopted a binary Gaussian example and provide Observation 4.1. Detailed discussions are deferred to Appendix A.3.

> *Observation* 4.1. Theoretically, we show the connection between error probability under the noisy data distribution and under the clean data distribution. Then, we provide insights on how **FR** mitigates the incurred bias term brought by the noisy data distribution.

Table 1: Performance comparisons on synthetic long-tailed noisy CIFAR datasets (noise type: imbalance-noise & symmetric noise), best-achieved averaged accuracy on a class-balanced test data are reported. Results in **bold**: **FR** improves the performance of the baseline methods, respectively.

| | CIFAR-10 ($\rho=0.2$) | | | CIFAR-10 ($\rho=0.5$) | | | CIFAR-100 ($\rho=0.2$) | | | CIFAR-100 ($\rho=0.5$) | | |
|---|---|---|---|---|---|---|---|---|---|---|---|---|
| **Noise type: Imbalance Noise** | | | | | | | | | | | | |
| Noise Ratio | | | | | | | | | | | | |
| Imbalance Ratio | $r=10$ | $r=50$ | $r=100$ | $r=10$ | $r=50$ | $r=100$ | $r=10$ | $r=50$ | $r=100$ | $r=10$ | $r=50$ | $r=100$ |
| CE | 79.75 | 65.98 | 60.03 | 65.38 | 47.51 | 37.44 | 46.02 | 31.44 | 26.98 | 29.58 | 16.93 | 13.87 |
| CE + FR (KNN) | **80.46** | **69.00** | **61.64** | 65.87 | 46.69 | **39.97** | **46.18** | 31.03 | **27.60** | **30.25** | 16.79 | **15.19** |
| CE + FR (G2) | **80.44** | **67.29** | **65.12** | **68.62** | **49.43** | **39.69** | **46.38** | **32.32** | **28.53** | **32.35** | **19.03** | **15.93** |
| LS (Lukasik et al., 2020) | 82.52 | 69.08 | 59.07 | 67.73 | 36.17 | 32.92 | 47.80 | 33.66 | 26.36 | 34.02 | 17.28 | 14.10 |
| LS + FR (KNN) | **82.78** | **70.06** | **59.27** | **68.99** | **36.55** | **36.63** | **48.27** | 33.01 | **27.60** | 32.01 | 17.14 | 14.07 |
| LS + FR (G2) | 82.02 | **70.24** | **60.33** | **70.50** | **44.11** | **35.49** | 47.30 | **33.86** | **29.67** | **34.51** | **17.84** | **16.68** |
| NLS (Wei et al., 2021b) | 79.91 | 65.98 | 58.82 | 64.74 | 41.01 | 34.16 | 46.05 | 31.48 | 27.09 | 29.86 | 16.84 | 13.87 |
| NLS + FR (KNN) | **80.17** | **68.61** | **62.88** | **68.65** | **47.42** | **36.79** | 45.72 | **32.25** | 27.01 | 28.85 | **17.23** | **14.18** |
| NLS + FR (G2) | **80.36** | **68.25** | **63.50** | **69.70** | **49.01** | **38.26** | 43.15 | **33.78** | **28.69** | **32.30** | **19.62** | **15.64** |
| Focal (Lin et al., 2017) | 76.24 | 64.16 | 57.68 | 62.40 | 40.25 | 34.56 | 43.63 | 29.10 | 24.88 | 26.93 | 14.45 | 12.57 |
| Focal + FR (KNN) | **77.54** | 62.97 | 57.24 | 61.47 | **42.28** | **37.04** | 42.44 | 28.90 | **25.14** | **28.34** | **16.02** | **13.27** |
| Focal + FR (G2) | **78.56** | **66.07** | 56.55 | **64.10** | **43.61** | **38.15** | **45.63** | **31.87** | **27.58** | **29.80** | **17.67** | **15.30** |
| PL (Liu & Guo, 2020) | 78.43 | 55.61 | 54.20 | 47.71 | 31.96 | 30.13 | 45.32 | 33.05 | 29.91 | 28.01 | 20.25 | 16.65 |
| PL + FR (KNN) | **79.50** | **65.37** | 53.36 | **51.82** | **35.68** | **30.16** | 44.89 | **33.12** | 28.63 | 27.66 | 19.79 | **17.72** |
| PL + FR (G2) | **78.79** | **66.16** | **54.39** | **50.72** | **33.22** | 28.01 | 44.78 | **33.35** | 29.51 | **29.82** | 20.15 | **16.81** |
| Logit-adj (Menon et al., 2021) | 82.09 | 73.23 | 68.18 | 68.30 | 51.51 | 42.17 | 47.28 | 33.11 | 29.47 | 30.92 | 17.97 | 14.68 |
| Logit-adj + FR (G2) | **82.92** | **75.67** | **72.20** | **73.72** | **55.09** | 40.85 | 41.21 | **35.39** | 28.84 | 27.57 | **18.93** | **15.44** |
| Logit-adj + FR (KNN) | **82.48** | **73.65** | **68.48** | **70.89** | 49.23 | **42.93** | **47.66** | **33.18** | **29.50** | **31.85** | 17.59 | **15.25** |
| **Noise type: Symmetric Noise** | | | | | | | | | | | | |
| Noise Ratio | | | | | | | | | | | | |
| Imbalance Ratio | $r=10$ | $r=50$ | $r=100$ | $r=10$ | $r=50$ | $r=100$ | $r=10$ | $r=50$ | $r=100$ | $r=10$ | $r=50$ | $r=100$ |
| CE | 80.70 | 65.04 | 61.80 | 70.48 | 51.53 | 36.44 | 46.02 | 30.93 | 26.98 | 29.93 | 16.70 | 4.76 |
| CE + FR (KNN) | **81.19** | **69.95** | **63.97** | **71.75** | **52.93** | **45.63** | **46.33** | 30.82 | **27.19** | **31.12** | **17.68** | **15.39** |
| CE + FR (G2) | **81.64** | **70.84** | **65.14** | **71.44** | **56.50** | **46.33** | **47.70** | **34.34** | **30.78** | **31.58** | **21.70** | **19.10** |
| LS (Lukasik et al., 2020) | 83.23 | 71.69 | 65.69 | 72.85 | 50.59 | 34.90 | 47.90 | 33.81 | 29.95 | 26.56 | 21.74 | 19.39 |
| LS + FR (KNN) | **83.28** | 70.64 | 60.91 | **73.92** | **53.01** | **43.48** | **49.05** | 33.40 | **30.05** | **34.86** | 20.73 | 19.10 |
| LS + FR (G2) | 82.22 | 70.85 | 62.43 | **74.59** | **54.15** | **44.77** | **48.16** | **34.08** | **30.69** | **36.40** | **22.06** | **20.10** |
| NLS (Wei et al., 2021b) | 80.79 | 66.22 | 61.47 | 70.11 | 50.57 | 36.55 | 46.11 | 31.14 | 27.32 | 30.51 | 17.16 | 5.18 |
| NLS + FR (KNN) | **81.08** | **69.29** | **63.58** | **70.27** | **54.86** | 36.50 | **48.20** | **35.03** | **28.29** | 28.87 | **19.10** | **6.65** |
| NLS + FR (G2) | **81.37** | **70.60** | **64.73** | **71.30** | **56.24** | **37.29** | **47.67** | **34.32** | **30.75** | 29.62 | **22.17** | **8.04** |
| Focal (Lin et al., 2017) | 77.77 | 61.54 | 56.02 | 67.20 | 43.12 | 38.20 | 35.93 | 23.23 | 21.84 | 27.31 | 16.18 | 14.71 |
| Focal + FR (KNN) | **78.03** | **64.57** | **56.77** | **67.87** | 41.89 | 36.34 | **42.79** | **30.17** | **25.08** | **28.22** | **16.37** | 14.50 |
| Focal + FR (G2) | **78.83** | **65.56** | **60.35** | **68.21** | **47.09** | **41.74** | **46.33** | **32.56** | **27.77** | **29.70** | **16.47** | **15.29** |
| PL (Liu & Guo, 2020) | 79.73 | 66.82 | 42.12 | 55.52 | 33.18 | 33.06 | 44.60 | 32.91 | 28.69 | 27.38 | 18.52 | 17.25 |
| PL + FR (KNN) | 79.42 | 64.91 | **58.80** | 53.86 | **38.41** | 32.71 | **45.60** | 32.32 | 28.34 | **27.63** | **18.86** | 16.48 |
| PL + FR (G2) | 79.37 | 66.71 | **58.98** | **55.68** | **38.08** | **33.52** | **46.83** | **33.17** | **29.67** | **28.12** | **19.48** | **17.62** |
| Logit-adj (Menon et al., 2021) | 80.50 | 62.42 | 50.28 | 60.38 | 32.45 | 27.32 | 46.50 | 29.24 | 23.80 | 28.79 | 12.65 | 9.22 |
| Logit-adj + FR (KNN) | **80.66** | 62.07 | **51.04** | **62.32** | 31.23 | 22.41 | **47.22** | **29.34** | **24.70** | **29.95** | 12.44 | **9.28** |
| Logit-adj + FR (G2) | **81.82** | **62.62** | **52.35** | **63.34** | 31.14 | 21.93 | **48.13** | **30.18** | **24.06** | **29.35** | 12.37 | **9.26** |

# 5 EXPERIMENTS

In this section, we verify the effectiveness of **FR** on the synthetic long-tailed noisy CIFAR datasets (Krizhevsky et al., 2009) and a real-world large-scale noisily labeled long-tailed dataset Clothing1M (Xiao et al., 2015).

## 5.1 EXPERIMENT DESIGNS ON SYNTHETIC NOISILY LABELED LONG-TAILED CIFAR DATASETS

We empirically test the performance of **FR** on CIFAR-10, and CIFAR-100 (Krizhevsky et al., 2009).

**Generation of Synthetic Long-Tailed Data with Noisy Labels** Note that the class information could be viewed as a special case of sub-populations, in this subsection, we treat classes as the natural separation of sub-populations and consider the class-imbalance experiment setting with noisy labels. For the balanced $K$-class classification task with $n$ samples per class, the synthetic long-tail setting assumes that $k-$th class has only $n/(r^{\frac{k-1}{K-1}})$ samples by referring to the ground-truth labels (Cui et al., 2019). We adopt two label-noise transition models below.

**Model 1 (Imb):** The entries of the noise transition matrix are given by $T_{i,j} := \mathbb{P}(\widetilde{Y} = j | Y = i, X = x)$: $T_{i,j}$ returns $1 - \rho$ if $i = j$; otherwise, $\frac{\mathbb{P}(Y=j)\cdot\rho}{1-\mathbb{P}(Y=i)}$. $\rho$ is viewed as the overall error/noise rate. The Imb noise model (Wei et al., 2021c) assumes that samples are more likely to be mislabeled as frequent ones in real-world situations.

**Model 2 (Sym):** The generation of the symmetric noisy dataset is adopted from (Kim et al., 2019), where it assumed that $T_{i,j} = \frac{\rho}{K-1}, \forall i \neq j$, indicating that any other classes $i \neq j$ has the same chance of being flipped to class $j$. The diagonal entry $T_{i,i}$ (chance of a correct label) becomes $1 - \rho$.

Figure 6: How **FR** improves per class test accuracy w.r.t. the baseline method on CIFAR-10. In each sub-figure, the $x$-axis indicates the accuracy of a baseline. $y$-axis denotes the performance of baseline when **FR** is introduced. Each dot denotes the test accuracy pair $(\text{Acc}_{\text{Method}}, \text{Acc}_{\text{Method+FR}})$ for each sub-population. The black line $y = x$ stands for the case that **FR** has no effects on a particular sub-population. The blue (red) dot above (below) the line shows the robust treatment has positive (negative) effect on this sub-population compared with CE.

For both noise settings, we test **FR** with noise rates $\rho \in \{0.2, 0.5\}$, meaning the proportion of wrong labels in the long-tailed training set is $0.2$ or $0.5$.

**Separation of $\mathcal{G}_i$**    We consider two kinds of sub-population separation methods for **FR**.

- **Separation with Clustering Methods:** $\forall x \in X$, the representation of feature $x$ is given by the representation extractor $\phi(x)$, where $\phi(\cdot) : X \to \mathbb{R}^d$ maps the feature $x$ to a $d$-dimensional representation vector. Given a distance metric DM (i.e., the Euclidean distance), the distance between two extracted representations $\phi(x_1), \phi(x_2)$ is $\text{DM}(\phi(x_1), \phi(x_2))$. The sub-population could be separated through clustering algorithms such as $k$-means ($k = N$ here). Admittedly, obtaining a good representation extractor is non-trivial, we want to highlight that the separation of sub-populations is not highly demanding on the quality of the representation extractor, and the focus is to perform fairness regularizations on varied features.

- **Separation Directly via Pre-Trained Models:** In this case, $\forall x \in X$, we adopt an (Image-Net) pre-trained model for the separation, i.e., such a feature extractor $\phi(\cdot)$ maps each $x$ into a $d = N$-dimensional representation vector so that all features are automatically categorized into $N$ sub-populations.

## 5.2 EXPERIMENT RESULTS ON CIFAR DATASETS

In Table 1, we empirically show how **FR** helps with improving the classifier's performance when complemented with several methods in robust losses as well as approaches in class-imbalanced learning, under synthetic class-imbalanced CIFAR datasets with noisy labels, including Cross-Entropy loss (CE), Label Smoothing (LS) (Lukasik et al., 2020), Negative Label Smoothing (NLS) (Wei et al., 2021b), Focal Loss (Lin et al., 2017), PeerLoss (PL) (Liu & Guo, 2020), and Logit-adjustment (Logit-adj) (Menon et al., 2021). We fix the same training samples and labels for all methods. More details are available in Appendix C.2.

For **FR**, we adopted the fixed $\lambda$ for all sub-populations. We consider two types of sub-population separation methods: (i) KNN clustering, which splits the extracted features into $K$ clusters, with $K$ being the number of classes; (ii) Generate the separation by referring to the direct prediction made by a (Image-Net) pre-trained model. In our experiments, this method separates features into a head and a tail sub-population, and the ratio w.r.t. the amount of samples between two sub-populations is $\approx 5$.

**Results**    In Table 1, we provide the baseline performance as well as the corresponding performances when **FR** is introduced. **FR** (KNN) denotes the scenario where we adopt the KNN clustering for sub-population separation, and the number of sub-populations is the same as the number of classes. We did not consider the noisy (class) labels as the sub-population index due to the fact that the noisy labels may contain the wrong ones. Empirically, we observe that **FR** (KNN) consistently improves the baseline methods on the class-imbalanced CIFAR-10 dataset, under the Imb and Sym noise. However, **FR** (KNN) could not improve significantly on the class-imbalanced CIFAR-100 dataset. One reason is that, in the batch update, the number of samples in each sub-population is too small (the average number is $128/100 = 1.28$), resulting in large variance in calculating **FR** as Eqn. (3). As an alternative, we report the performance of **FR** (G2) as well, where samples are categorized into 2 sub-populations by the (Image-Net) pre-trained model. Surprisingly, **FR** (G2) improves the performance of 6 baselines in most settings, as highlighted in Table 1. Constraining the classifier to have relative fairness performances is beneficial when learning with noisy and long-tailed data.

We further adopt the CIFAR-10 dataset ($\rho = 0.5, r = 50$) and visualize how **FR** influences the per-class accuracy by referring to the performance of each baseline. Each blue point in Figure 6 indicates the scenario where **FR** improves the test accuracy of a class over the corresponding baseline. Points in the lower left corner (where tail populations are usually located) further illustrate that **FR** consistently improves the performance of tail sub-populations.

**Hypothesis Testing w.r.t. FR**   We adopt paired student t-test to verify the conclusion that **FR** helps with improving the test accuracy. In Table 2, positive statistics indicate that the **FR** generally improves the performance (test accuracy) of the baseline method. The $p$-value that is smaller than 0.1 means there exist significant differences between the two accuracy lists. In such scenarios, we should reject the null hypothesis and adopt the alternative hypothesis. Table 2 shows that **FR** (G2) brings significant performance improvements in most settings (5/6 in CIFAR-10 and 6/6 in CIFAR-100),

Table 2: Paired student t-test results w.r.t. the effectiveness of **FR**. Rows marked with "$\sqrt{}$" mean **FR** improve the performance of the baseline methods significantly ($p$-value satisfies that $p < 0.1$ and the statistics is positive); "−" indicates there exist no significant differences after adopting **FR**.

| Method | FR Type | CIFAR-10 | | | CIFAR-100 | | |
|---|---|---|---|---|---|---|---|
| | | statistics | $p$-value | Better | statistics | $p$-value | Better |
| CE | FR (KNN) | 2.962 | 0.013 | $\sqrt{}$ | 1.489 | 0.165 | − |
| CE | FR (G2) | 4.313 | 0.001 | $\sqrt{}$ | 3.083 | 0.010 | $\sqrt{}$ |
| LS | FR (KNN) | 1.214 | 0.250 | − | 0.748 | 0.470 | − |
| LS | FR (G2) | 1.851 | 0.091 | $\sqrt{}$ | 1.926 | 0.080 | $\sqrt{}$ |
| NLS | FR (KNN) | 4.235 | 0.001 | $\sqrt{}$ | 1.692 | 0.119 | − |
| NLS | FR (G2) | 4.909 | <0.000 | $\sqrt{}$ | 3.237 | 0.008 | $\sqrt{}$ |
| PL | FR (KNN) | 1.859 | 0.090 | $\sqrt{}$ | -0.620 | 0.548 | − |
| PL | FR (G2) | 1.847 | 0.092 | $\sqrt{}$ | 2.345 | 0.039 | $\sqrt{}$ |
| Focal | FR (KNN) | 0.886 | 0.395 | − | 2.218 | 0.049 | $\sqrt{}$ |
| Focal | FR (G2) | 5.249 | <0.000 | $\sqrt{}$ | 4.105 | 0.002 | $\sqrt{}$ |
| Logit-adj | FR (KNN) | 1.171 | 0.266 | − | -0.419 | 0.684 | − |
| Logit-adj | FR (G2) | 0.255 | 0.803 | − | 2.410 | 0.035 | $\sqrt{}$ |

indicating the effectiveness of our method. Besides, **FR** (KNN) shows significant performance improvements only in several settings (but there are still improvements in most cases), which can be explained by our previous discussion that a large number of sub-populations may make the learning unstable. More details appear in Appendix C.2.

## 5.3   EXPERIMENT RESULTS ON CLOTHING1M DATASET

Clothing1M is a large-scale feature-dependent human-level noisy clothes dataset. We adopt the same baselines as reported in CIFAR experiments for Clothing1M. More detailed descriptions are given in Appendix C.3.

We try implementing **FR** with different $\lambda$ chosen from the set $\{0.0, 0.1, 0.2, 0.4, 0.6, 0.8, 1.0, 2.0\}$, where $\lambda = 0.0$ indicates the training of baseline methods without **FR**. In Table 3, the default setting of **FR** ($\lambda = 1.0$) consistently reaches competitive performances by comparing to other $\lambda$s, except for the experiments w.r.t. NLS.

Table 3: Performance comparisons on real-world imbalanced noisily labeled dataset (Clothing1M), best and last-epoch achieved test accuracy are reported. Results in **bold** mean **FR** improves the performance of the baseline methods, respectively. Performances of **FR** with different $\lambda$s are provided.

| Method | $\lambda$ | 0.0 | 0.1 | 0.2 | 0.4 | 0.6 | 0.8 | 1.0 | 2.0 |
|---|---|---|---|---|---|---|---|---|---|
| CE | Best | 72.68 | 72.44 | **72.93** | **72.74** | **73.10** | **72.80** | **72.99** | 72.45 |
| | Last | 72.22 | 71.99 | **72.25** | **72.24** | **72.51** | **72.53** | **72.58** | 72.20 |
| LS | Best | 72.55 | **72.71** | **72.69** | 72.34 | 72.41 | 72.44 | **72.70** | **72.56** |
| | Last | 72.03 | **72.11** | **72.14** | **72.12** | **72.12** | **72.06** | **72.33** | **72.24** |
| NLS | Best | 74.46 | **74.48** | **74.47** | **74.49** | **74.48** | **74.49** | **74.49** | **74.50** |
| | Last | 74.00 | 73.99 | 73.97 | 73.98 | 73.98 | 73.97 | 73.97 | 73.97 |
| PL | Best | 73.00 | **73.27** | **73.13** | **73.15** | **73.13** | **73.22** | **73.08** | **73.02** |
| | Last | 72.73 | **72.91** | **72.87** | 72.69 | **72.76** | **73.12** | 72.71 | 72.69 |
| Focal | Best | 72.71 | 72.60 | **72.71** | 72.60 | **72.92** | 72.66 | **72.91** | 72.46 |
| | Last | 72.16 | **72.21** | 72.04 | **72.18** | **72.30** | **72.36** | **72.51** | **72.46** |
| Logit-adj | Best | 72.43 | **72.52** | **72.48** | 71.88 | 72.22 | **72.45** | **72.67** | 72.06 |
| | Last | 72.22 | 72.15 | 72.14 | 71.58 | 71.83 | 71.94 | **72.23** | 71.92 |

to other $\lambda$s. Besides, we observe that most positive $\lambda$s that are close to $\lambda = 1.0$ tend to have better performances than those close to $\lambda = 0.0$, indicating the effectiveness as well as hyper-parameter in-sensitiveness of the introduced fairness regularizer.

## 6   CONCLUSIONS

In this paper, we qualitatively and quantitatively analyzed the influence of sub-populations under various metrics, where we observed disparate impacts incurred by sub-populations, especially when the label noise presents. What is more, our experiment results also reveal that existing robust solutions improve the performance of certain sub-populations at the cost of hurting others, hence leading to unfair performances among sub-populations. We then propose **F**airness **R**egularizer (**FR**), which encourages the learned classifier to achieve fair performances across sub-populations. Extensive experiment results demonstrate the effectiveness of **FR**, indicating that fairness constraints improve the learning from noisily labeled long-tailed data. One limitation is that our proposed method has only been tested on image classification tasks. The performance on other tasks needs more exploration. We defer more detailed discussions to the beginning of the Appendix.

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

# APPENDIX

**Broader Impacts** This paper considers the setting where the classification task observes noisy annotations and imbalanced clean label priors. Most prior works treat either population bias or label noise in an isolated way and do not explicitly consider the coupling effects of the two. However, in practical cases, real-world data is noisy and imbalanced. Our proposed Fairness Regularizer (FR) addresses this practical case and can be viewed as a plug-in item to extend existing solutions easily.

**Code of Ethics** There is no sensitive attribute in our method and results. Therefore, the potential negative impacts do not apply to our work. Our code is uploaded along with other supplementary materials.

**Limitations** The proposed method has only been tested on the image classification tasks. The performance on other tasks (i.e., text classification) needs more exploration.

Before presenting more materials, we find it necessary to clarify potential misunderstandings.

**Clarification** Throughout this work, the saying of groups is a generalized definition of the separation of samples, which includes many popular settings as special cases, i.e.,

- Class-relevant: the class name is actually a natural separation of samples, such separations could be more fine-grained class-related (such as further splitting the class "cat" into finer separations by referring to the breed of cats);

- Class-irrelevant: such population could also be class-irrelevant, for example, in image classification tasks where the gender information is the (hidden) attribute information of each image while the class/label does not disclose this information.

**Organization** The rest of the Appendix is organized as follows.

- Section A theoretically demonstrates why special treatments on sub-populations are necessary, and why **F**airness **R**egularizer (**FR**) improves learning from the noisily labeled long-tailed data.

- Section B includes all omitted proofs for theoretical conclusions.

- Section C gives additional experiment details and results.

- Section D gives detailed discussions/comparisons with several more recent related works.

## A  LONG-TAILED SUB-POPULATIONS DESERVE SPECIAL TREATMENTS

In light of the empirical observations, we now theoretically explore the impacts of sub-populations when learning with long-tailed noisy data, through a binary Gaussian example. Note that classes could be viewed as a special case of sub-populations, we adopt the class-level long-tailed distributions for illustration.

### A.1  FORMULATION

Consider the binary classification task such that $K = 2$, and the data samples are generated by $P_{XY}$, which is the mixture of two Gaussians. Suppose $X^{\pm} := (X|Y = \pm 1) \sim \mathcal{N}(\mu_{\pm}, \sigma^2)$ where $\mathcal{N}$ is the Gaussian distribution, and $\mathbb{P}(Y = +1) = \mathbb{P}(Y = -1)$. W.l.o.g., we assume that $\mu_+ > \mu_-$. Suppose a classifier $f$ was trained on the noisy (and potentially imbalanced/long-tailed) training data $X_{\mathrm{I}} := \{x_i\}_{i=1}^n$ where the corresponding noisy label of $x_i$ is $\tilde{y}_i \in \widetilde{Y}$, $\forall i \in [n]$. Samples $x_i$ were drawn non-uniformly (i.e., imbalance) from $X$, and the ground-truth label of samples $x_i$ is $y_i$ given by $Y$.

To inspect on the influence of sub-populations, we further split the imbalanced noisily labeled training data into two parts by referring to their clean labels: head-Gaussian data and tail-Gaussian data. For $x_i \sim X^{\pm}, y \in \{\pm 1\}$, we denote the set of head and tail data in class $\pm 1$ as $S_{\mathrm{H}}^{\pm} := X_{\mathrm{I}} \cap X_{\mathrm{H}}^{\pm}, S_{\mathrm{T}}^{\pm} := X_{\mathrm{I}} \cap X_{\mathrm{T}}^{\pm}$, where $X_{\mathrm{H}}^{\pm} := \{x \sim X^{\pm} | \frac{x - \mu_{\pm}}{\sigma} \cdot y \geq -\eta\}, X_{\mathrm{T}}^{\pm} := \{x \sim X^{\pm} | \frac{x - \mu_{\pm}}{\sigma} \cdot y < -\eta\}$ respectively.

To clarify, replacing all "$\pm$" symbols by $+1$ will return the notation for class $+1$. And $G \in \{\mathrm{H}, \mathrm{T}\}$ in this setting.

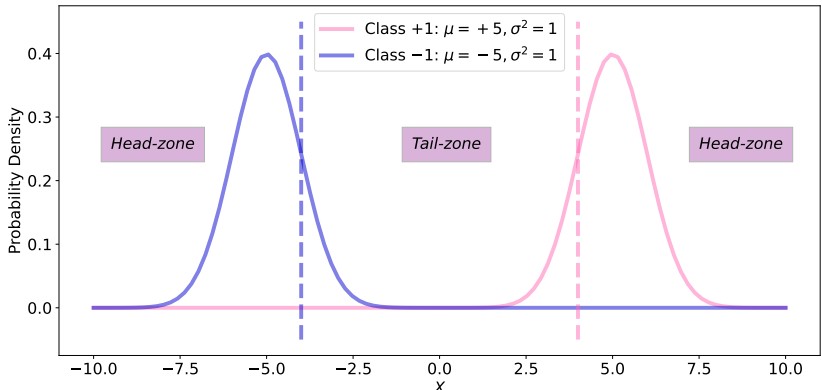

Figure 7: An illustration of head/tail separations: when $\mu_+ = +5, \mu_- = -5, \sigma^2 = 1, \eta = 1$, the probability density distribution of Class +1 (Pink) and Class -1 (Blue) are drawn. $x$-axis indicates the Gaussian samples drawn from two Gaussian distributions, $y$-axis is the corresponding probability density of samples being equal to $x$.

In the view of sub-populations, we assume that the noise transition differs w.r.t. the head and tail proportion, since tail populations are more misleading in the classification (i.e., in Figure 7, the label of samples in the "tail-zone" is more likely to be wrongly given). Assume that the noise transition matrix in the head samples and tail samples follow $T_\mathrm{H}, T_\mathrm{T}$ respectively:

$$T_\mathrm{H} = \begin{pmatrix} 1 - \rho_\mathrm{H}^- & \rho_\mathrm{H}^- \\ \rho_\mathrm{H}^+ & 1 - \rho_\mathrm{H}^+ \end{pmatrix}, \quad T_\mathrm{T} = \begin{pmatrix} 1 - \rho_\mathrm{T}^- & \rho_\mathrm{T}^- \\ \rho_\mathrm{T}^+ & 1 - \rho_\mathrm{T}^+ \end{pmatrix}.$$

To refer to the noisy labels, we add the $\widetilde{\cdot}$ sign for the notations that are w.r.t. $\widetilde{Y}$ instead of $Y$: i.e., $\widetilde{X^\pm} := (X|\widetilde{Y} = \pm1)$, $\widetilde{S_\mathrm{H}^\pm} := X_\mathrm{I} \cap \widetilde{X_\mathrm{H}^\pm}$, with $\widetilde{X_\mathrm{H}^\pm}$ denoting the noisy data distribution such that the clean data distribution belongs to the head subpopulation (either $X_\mathrm{H}^+$ or $X_\mathrm{H}^-$), and its noisy label is $\pm$. Similarly, $\widetilde{S_\mathrm{T}^\pm} := X_\mathrm{I} \cap \widetilde{X_\mathrm{T}^\pm}$, with $\widetilde{X_\mathrm{T}^\pm}$ denoting the noisy data distribution such that the clean data distribution belongs to the tail subpopulation (either $X_\mathrm{T}^+$ or $X_\mathrm{T}^-$), and its noisy label is $\pm$.

**An example** If $\rho_\mathrm{H}^+ = 0.2, \rho_\mathrm{H}^- = 0.3$,

- With probability 0.2, the sample $x_0$ drawn from $X_\mathrm{H}^+$ will flip its label from class $+$ to class $-$; and with probability 0.8, the sample $x_0$ drawn from $X_\mathrm{H}^+$ will keep its label (class $+$) unchanged.
- With probability 0.3, the sample $x_1$ drawn from $X_\mathrm{H}^-$ will flip its label from class $-$ to class $+$; and with probability 0.7, the sample $x_1$ drawn from $X_\mathrm{H}^-$ will keep its label (class $-$) unchanged.

W.l.o.g., we assume that the noise rates are not too large, i.e., $\rho_\mathrm{H}^\pm, \rho_\mathrm{T}^\pm \in [0.0.5)$. Besides, we are interested in the scenario where the ground-truth samples are imbalanced. And we can assume that the imbalance ratio $r := \frac{|S_\mathrm{H}^+| + |S_\mathrm{T}^+|}{|S_\mathrm{H}^-| + |S_\mathrm{T}^-|}$ satisfies $r > 1$.

## A.2 THE ERROR PROBABILITY

Given a classifier $f$, the *Error probability* is defined as the percentage of error rates under a given data distribution:

**Definition A.1** (Error probability). The error probability of a classifier $f$ on the data distribution $(X, Y)$ is defined as $\mathrm{Err}_X(f) := \mathbb{P}_{(X,Y)}(f(X) \neq Y)$.

Denote by $\Phi$ the cumulative distribution function (CDF) of the standard Gaussian distribution $\mathcal{N}(0,1)$, we derive the error probability for the four populations as:

**Proposition A.2.** *For any linear classifier of the form $f(x) = sign(x - \theta)$, we have:*

$$Err_{X_T^\pm}(f) - Err_{X_H^\pm}(f) \quad \propto \quad \Phi\left((\theta - \mu_\pm) \cdot (\pm 1)\right) \cdot sign\left((\mu_\pm - \theta) \cdot (\pm 1) - \eta\sigma\right).$$

$\mu_\pm$ denotes the mean of two Gaussians and the Bayes optimal classifier adopts the threshold $\theta^* := \frac{\mu_- + \mu_+}{2}$. We take the tail class $-1$ (replace all symbols "$\pm$" by "$-$") as an illustration:

- When $\theta \geq \mu_- + \eta\sigma$, the error probability gap $Err_{X_T^\pm}(f) - Err_{X_H^\pm}(f)$ is monotonically increasing w.r.t. the increase of $\Phi\left((\theta - \mu_-)\right) \cdot sign\left((\theta - \mu_-) - \eta\sigma\right) = \Phi\left((\theta - \mu_-)\right)$. Without additional treatments, the classifier over-fits on the head class to achieve a lower error probability. As a result, $\theta$ decreases, and the error gap between two populations in the tail class enlarges.

- When $\theta$ decreases small enough, i.e., $\theta < \mu_- + \eta\sigma$, the error probability gap $Err_{X_T^\pm}(f) - Err_{X_H^\pm}(f)$ is monotonically increasing w.r.t. the increase of $-\Phi\left((\theta - \mu_-)\right)$. Further decreasing $\theta$ will make both populations in the tail class yield a large error probability.

## A.3 WHY DOES **FR** HELP WITH IMPROVEMENTS

Building upon the previous discussions, to show why **FR** helps with improving the learning, we first derive the per sub-population error probability w.r.t. the noisy labels, since in practice, clean labels are not available for the **FR** to constrain.

**Lemma A.3.** *The error probability of a classifier $f$ on the per-population noisy data distribution $(X, \widetilde{Y})$ could be expressed in the forms of error probabilities under the clean data distribution, specifically:*

$$Err_{\widetilde{X_H^+}}(f) = p \cdot (1 - \rho_H^+) \cdot Err_{X_H^+} + (1-p) \cdot \rho_H^- \cdot (1 - Err_{X_H^-});$$

$$Err_{\widetilde{X_H^-}}(f) = p \cdot \rho_H^+ \cdot Err_{X_H^+} + (1-p) \cdot (1 - \rho_H^-) \cdot (1 - Err_{X_H^-});$$

$$Err_{\widetilde{X_T^+}}(f) = p \cdot (1 - \rho_T^+) \cdot Err_{X_T^+} + (1-p) \cdot \rho_T^- \cdot (1 - Err_{X_T^-});$$

$$Err_{\widetilde{X_T^-}}(f) = p \cdot \rho_T^+ \cdot Err_{X_T^+} + (1-p) \cdot (1 - \rho_T^-) \cdot (1 - Err_{X_T^-}).$$

Although the overall error probability on the clean data distribution is:

$$\text{Err}(f) := \mathbb{P}(X_H^+) \cdot \text{Err}_{X_H^+}(f) + \mathbb{P}(X_H^-) \cdot \text{Err}_{X_H^-}(f) + \mathbb{P}(X_T^+) \cdot \text{Err}_{X_T^+}(f) + \mathbb{P}(X_T^-) \cdot \text{Err}_{X_T^-}(f).$$

when learning with noisy data distribution with imbalanced sub-populations, the optimal $f$ w.r.t. the noisy data distribution is supposed to be given by the optimum of the following Risk Minimization:

$$\text{RM:} \quad \min_f \quad \widetilde{\text{Err}}(f) := \mathbb{P}(\widetilde{X_H^+}) \cdot \text{Err}_{\widetilde{X_H^+}}(f) + \mathbb{P}(\widetilde{X_H^-}) \cdot \text{Err}_{\widetilde{X_H^-}}(f) + \mathbb{P}(\widetilde{X_T^+}) \cdot \text{Err}_{\widetilde{X_T^+}}(f) + \mathbb{P}(\widetilde{X_T^-}) \cdot \text{Err}_{\widetilde{X_T^-}}(f).$$

To distinguish the overall error probability under noisy and clean data distribution, we offer Theorem A.4:

**Theorem A.4.** *When $\mathbb{P}(Y = +1) = \mathbb{P}(Y = -1)$, an equivalent form of the minimization w.r.t. $\widetilde{Err}(f)$ is characterized by:*

$$\min_f \quad \widetilde{Err}(f) \iff \min_f \quad Err(f) - 2r \cdot \rho_H \cdot \left(Err_{\widetilde{X_H^+}}(f) - Err_{\widetilde{X_H^-}}(f)\right) - \rho_T \cdot \left(Err_{\widetilde{X_T^+}}(f) - Err_{\widetilde{X_T^-}}(f)\right),$$

*where we define the noise rate gaps as $\rho_H := \rho_H^+ - \rho_H^-, \rho_T := \rho_T^+ - \rho_T^-$, and the sub-population imbalance ratio as $r := \frac{1 - \Phi(-\eta)}{\Phi(-\eta)}$.*

Thus, constraining the classifier to perform fair performances (i.e., same training error probabilities such as $\text{Err}_{\widetilde{X_H^+}}(f) = \text{Err}_{\widetilde{X_H^-}}(f)$, and $\text{Err}_{\widetilde{X_T^+}}(f) = \text{Err}_{\widetilde{X_T^-}}(f)$), the optimal classifier training on the noisy data distribution with fairness regularizer yields the optimal classifier by refer to the clean data distribution! We then have:

**Corollary A.5.** *When* $\mathbb{P}(Y = +1) = \mathbb{P}(Y = -1)$, ***FR*** *constrains the error probability (performance gap) between* $\widetilde{X_H^+}$ *and* $\widetilde{X_H^-}$, $\widetilde{X_T^+}$ *and* $\widetilde{X_T^-}$. *As a result, we have*

$$\min_f \quad \widetilde{Err}(f) \qquad s.t. \qquad Err_{\widetilde{X_H^+}}(f) = Err_{\widetilde{X_H^-}}(f), Err_{\widetilde{X_T^+}}(f) = Err_{\widetilde{X_T^-}}(f)$$

$$\iff \quad \min_f \quad Err(f) \qquad s.t. \qquad Err_{\widetilde{X_H^+}}(f) = Err_{\widetilde{X_H^-}}(f), Err_{\widetilde{X_T^+}}(f) = Err_{\widetilde{X_T^-}}(f).$$

The proof is straightforward from the result in Theorem A.4.

## B    OMITTED PROOFS

### B.1    PROOF OF PROPOSITION A.2

*Proof.* For the head population in Class $+1$, we can derive the error probability as:

$$\text{Err}_{X_H^+}(f) := \mathbb{P}_{(X_H^+, Y)}(f(X_H^+) \neq Y) = \mathbb{P}_{(X_H^+, Y)}\big((X_H^+ - \theta)Y < 0\big) = \mathbb{P}_{(X_H^+, Y)}\big(X_H^+ < \theta\big).$$

Due to the separation of head and tail in Class $+1$, we then have:

$$\begin{aligned}
\text{Err}_{X_H^+}(f) &= \frac{\mathbb{P}_{x \sim \mathcal{N}(\mu_+, \sigma^2)}\big(x < \theta, \frac{x - \mu_+}{\sigma} \geq -\eta\big)}{\mathbb{P}_{x \sim \mathcal{N}(\mu_+, \sigma^2)}\big(\frac{x - \mu_+}{\sigma} \geq -\eta\big)} \\
&= \frac{\mathbb{P}\big(\mathcal{N}(\mu_+, \sigma^2) < \theta, \mathcal{N}(0, \sigma^2) \geq -\eta\sigma\big)}{\mathbb{P}(\mathcal{N}(0, 1) \geq -\eta)} \\
&= \frac{\mathbb{P}\big(\mathcal{N}(0, 1) < \frac{\theta - \mu_+}{\sigma}, \mathcal{N}(0, 1) \geq -\eta\big)}{1 - \Phi(-\eta)}.
\end{aligned}$$

where we denote by $\Phi$ the CDF of the standard Gaussian distribution $\mathcal{N}(0, 1)$, and $\Phi(a) = 1 - \Phi(-a)$. Similarly, for the tail population in Class $+1$, we can derive the error probability as:

$$\text{Err}_{X_T^+}(f) := \mathbb{P}_{(X_T^+, Y)}(f(X_T^+) \neq Y) = \mathbb{P}_{(X_T^+, Y)}\big((X_T^+ - \theta)Y < 0\big) = \mathbb{P}_{(X_T^+, Y)}\big(X_T^+ < \theta\big)$$

$$= \frac{\mathbb{P}_{x \sim \mathcal{N}(\mu_+, \sigma^2)}\big(x < \theta, \frac{x - \mu_+}{\sigma} < -\eta\big)}{\mathbb{P}_{x \sim \mathcal{N}(\mu_+, \sigma^2)}\big(\frac{x - \mu_+}{\sigma} < -\eta\big)}$$

$$= \frac{\mathbb{P}\big(\mathcal{N}(\mu_+, \sigma^2) < \theta, \mathcal{N}(0, \sigma^2) < -\eta\sigma\big)}{\mathbb{P}(\mathcal{N}(0, 1) < -\eta)}$$

$$= \frac{\mathbb{P}\big(\mathcal{N}(0, 1) < \frac{\theta - \mu_+}{\sigma}, \mathcal{N}(0, 1) < -\eta\big)}{\Phi(-\eta)}.$$

For the populations in Class $-1$, we have:

$$\text{Err}_{X_H^-}(f) := \mathbb{P}_{(X_H^-, Y)}(f(X_H^-) \neq Y) = \mathbb{P}_{(X_H^-, Y)}\big((X_H^- - \theta)Y > 0\big) = \mathbb{P}_{(X_H^-, Y)}\big(X_H^- > \theta\big)$$

$$= \frac{\mathbb{P}_{x \sim \mathcal{N}(\mu_-, \sigma^2)}\big(x > \theta, \frac{x - \mu_-}{\sigma} \leq \eta\big)}{\mathbb{P}_{x \sim \mathcal{N}(\mu_-, \sigma^2)}\big(\frac{x - \mu_-}{\sigma} \leq \eta\big)}$$

$$= \frac{\mathbb{P}\big(\mathcal{N}(\mu_-, \sigma^2) > \theta, \mathcal{N}(0, \sigma^2) \leq \eta\sigma\big)}{\mathbb{P}(\mathcal{N}(0, 1) \leq \eta)}$$

$$= \frac{\mathbb{P}\big(\mathcal{N}(0, 1) > \frac{\theta - \mu_-}{\sigma}, \mathcal{N}(0, 1) \leq \eta\big)}{1 - \Phi(-\eta)}.$$

$$\text{Err}_{X_{\text{T}}^-}(f) := \mathbb{P}_{(X_{\text{T}}^-, Y)}(f(X_{\text{T}}^-) \neq Y)$$

$$= \mathbb{P}_{(X_{\text{T}}^-, Y)}\big((X_{\text{T}}^- - \theta)Y > 0\big)$$

$$= \mathbb{P}_{(X_{\text{T}}^-, Y)}\big(X_{\text{T}}^- > \theta\big)$$

$$= \frac{\mathbb{P}_{x \sim \mathcal{N}(\mu_-, \sigma^2)}\big(x > \theta, \frac{x - \mu_-}{\sigma} > \eta\big)}{\mathbb{P}_{x \sim \mathcal{N}(\mu_-, \sigma^2)}\big(\frac{x - \mu_-}{\sigma} > \eta\big)}$$

$$= \frac{\mathbb{P}\big(\mathcal{N}(\mu_-, \sigma^2) > \theta, \mathcal{N}(0, \sigma^2) > \eta\sigma\big)}{\mathbb{P}(\mathcal{N}(0, 1) > \eta)}$$

$$= \frac{\mathbb{P}\big(\mathcal{N}(0, 1) > \frac{\theta - \mu_-}{\sigma}, \mathcal{N}(0, 1) > \eta\big)}{\Phi(-\eta)}.$$

The above thresholds can be further simplified into following forms given the cumulative distribution function (CDF) of the normal Gaussian distribution. If $\theta \leq \mu_+ - \eta\sigma$, then:

$$\text{Err}_{X_{\text{H}}^+}(f) = \frac{0}{1 - \Phi(-\eta)} = 0, \quad \text{Err}_{X_{\text{T}}^+}(f) = \frac{\mathbb{P}\big(\mathcal{N}(0, 1) < \min(\frac{\theta - \mu_+}{\sigma}, -\eta)\big)}{\Phi(-\eta)} = \frac{\Phi\big(\frac{\theta - \mu_+}{\sigma}\big)}{\Phi(-\eta)};$$

otherwise, we have $\theta > \mu_+ - \eta\sigma$ and:

$$\text{Err}_{X_{\text{H}}^+}(f) = \frac{\mathbb{P}\big(-\eta \leq \mathcal{N}(0, 1) < \frac{\theta - \mu_+}{\sigma}\big)}{1 - \Phi(-\eta)} = \frac{\Phi(\frac{\theta - \mu_+}{\sigma}) - \Phi(-\eta)}{1 - \Phi(-\eta)}, \quad \text{Err}_{X_{\text{T}}^+}(f) = 1.$$

As for the class $-1$, when $\theta \geq \mu_- + \eta\sigma$, we obtain:

$$\text{Err}_{X_{\text{H}}^-}(f) = \frac{0}{1 - \Phi(-\eta)} = 0, \quad \text{Err}_{X_{\text{T}}^-}(f) = \frac{\mathbb{P}\big(\mathcal{N}(0, 1) > \max(\frac{\theta - \mu_-}{\sigma}, \eta)\big)}{\Phi(-\eta)} = \frac{\Phi\big(\frac{\mu_- - \theta}{\sigma}\big)}{\Phi(-\eta)};$$

otherwise, we have $\theta < \mu_- + \eta\sigma$ and:

$$\text{Err}_{X_{\text{H}}^-}(f) = \frac{\mathbb{P}\big(\frac{\theta - \mu_-}{\sigma} < \mathcal{N}(0, 1) \leq \eta\big)}{1 - \Phi(-\eta)} = \frac{\Phi(\frac{\mu_- - \theta}{\sigma}) - \Phi(-\eta)}{1 - \Phi(-\eta)}, \quad \text{Err}_{X_{\text{T}}^-}(f) = 1.$$

We take Class $-1$ for illustration, when $\theta \geq \mu_- + \eta\sigma$, we have: $(\mu_- - \theta) \cdot (-1) - \eta\sigma \geq 0$. In this case, the difference of error probabilities in tail and head populations becomes:

$$\text{Err}_{X_{\text{T}}^-}(f) - \text{Err}_{X_{\text{H}}^-}(f) = \frac{\Phi\big(\frac{\mu_- - \theta}{\sigma}\big)}{\Phi(-\eta)} \propto \Phi\big(\frac{\mu_- - \theta}{\sigma}\big) \propto \Phi(\mu_- - \theta)$$

$$\propto \quad \Phi\big((\theta - \mu_-) \cdot (-1)\big) \cdot \text{sign}\big((\mu_- - \theta) \cdot (-1) - \eta\sigma\big).$$

When $\theta < \mu_- + \eta\sigma$, we have: $(\mu_- - \theta) \cdot (-1) - \eta\sigma < 0$. In this case, the difference of error probabilities in tail and head populations becomes:

$$\text{Err}_{X_{\text{T}}^-}(f) - \text{Err}_{X_{\text{H}}^-}(f) = 1 - \frac{\Phi(\frac{\mu_- - \theta}{\sigma}) - \Phi(-\eta)}{1 - \Phi(-\eta)}$$

$$= \frac{1 - \Phi(\frac{\mu_- - \theta}{\sigma})}{1 - \Phi(-\eta)}$$

$$= \frac{\Phi(\frac{\theta - \mu_-}{\sigma})}{1 - \Phi(-\eta)} \propto \Phi\big(\frac{\theta - \mu_-}{\sigma}\big) \propto \Phi(\mu_- - \theta) \cdot (-1)$$

$$\propto \quad \Phi\big((\theta - \mu_-) \cdot (-1)\big) \cdot \text{sign}\big((\mu_- - \theta) \cdot (-1) - \eta\sigma\big).$$

For Class $+1$, the conclusion could be derived similarly.

$\square$

## B.2 PROOF OF LEMMA A.3

*Proof.* Note that:

$$
\mathrm{Err}_{X_{\mathrm{H}}^{+}}(f) = \frac{\mathbb{P}\big(\mathcal{N}(0,1) < \frac{\theta-\mu_{+}}{\sigma}, \mathcal{N}(0,1) \geq -\eta\big)}{1 - \Phi(-\eta)}, \qquad
\mathrm{Err}_{X_{\mathrm{T}}^{+}}(f) = \frac{\mathbb{P}\big(\mathcal{N}(0,1) < \frac{\theta-\mu_{+}}{\sigma}, \mathcal{N}(0,1) < -\eta\big)}{\Phi(-\eta)}.
$$

$$
\mathrm{Err}_{X_{\mathrm{H}}^{-}}(f) = \frac{\mathbb{P}\big(\mathcal{N}(0,1) > \frac{\theta-\mu_{-}}{\sigma}, \mathcal{N}(0,1) \leq \eta\big)}{1 - \Phi(-\eta)}, \qquad
\mathrm{Err}_{X_{\mathrm{T}}^{-}}(f) = \frac{\mathbb{P}\big(\mathcal{N}(0,1) > \frac{\theta-\mu_{-}}{\sigma}, \mathcal{N}(0,1) > \eta\big)}{\Phi(-\eta)}.
$$

Thus, denote by $p := \mathbb{P}(Y = +)$, we have:

$$
\begin{aligned}
\mathrm{Err}_{\widetilde{X_{\mathrm{H}}^{+}}}(f) &= \mathbb{P}_{(\widetilde{X_{\mathrm{H}}^{+}}, \widetilde{Y})}(f(\widetilde{X_{\mathrm{H}}^{+}}) \neq \widetilde{Y}) \\
&= \mathbb{P}(\widetilde{Y} = +, Y = +|X_{\mathrm{H}}^{+}) \cdot \mathbb{P}_{(\widetilde{X_{\mathrm{H}}^{+}}, \widetilde{Y})}\big((\widetilde{X_{\mathrm{H}}^{+}} - \theta)Y < 0\big) + \mathbb{P}(\widetilde{Y} = +, Y = -|X_{\mathrm{H}}^{-}) \cdot \mathbb{P}_{(\widetilde{X_{\mathrm{H}}^{+}}, \widetilde{Y})}\big((\widetilde{X_{\mathrm{H}}^{+}} - \theta)Y > 0\big) \\
&= p \cdot (1 - \rho_{\mathrm{H}}^{+}) \cdot \mathbb{P}_{(X_{\mathrm{H}}^{+}, Y=+)}\big((X_{\mathrm{H}}^{+} - \theta)Y < 0\big) + (1 - p) \cdot \rho_{\mathrm{H}}^{-} \cdot \mathbb{P}_{(X_{\mathrm{H}}^{-}, Y=-)}\big((X_{\mathrm{H}}^{-} - \theta)Y > 0\big) \\
&= p \cdot (1 - \rho_{\mathrm{H}}^{+}) \cdot \mathbb{P}_{(X_{\mathrm{H}}^{+}, Y=+)}\big(X_{\mathrm{H}}^{+} < \theta\big) + (1 - p) \cdot \rho_{\mathrm{H}}^{-} \cdot \mathbb{P}_{(X_{\mathrm{H}}^{-}, Y=-)}\big(X_{\mathrm{H}}^{-} < \theta\big) \\
&= p \cdot (1 - \rho_{\mathrm{H}}^{+}) \cdot \mathrm{Err}_{X_{\mathrm{H}}^{+}} + (1 - p) \cdot \rho_{\mathrm{H}}^{-} \cdot (1 - \mathrm{Err}_{X_{\mathrm{H}}^{-}}).
\end{aligned}
$$

Similarly, we could derive:

$$
\begin{aligned}
\mathrm{Err}_{\widetilde{X_{\mathrm{H}}^{-}}}(f) &= \mathbb{P}_{(\widetilde{X_{\mathrm{H}}^{-}}, \widetilde{Y})}(f(\widetilde{X_{\mathrm{H}}^{-}}) \neq \widetilde{Y}) \\
&= \mathbb{P}(\widetilde{Y} = -, Y = +|X_{\mathrm{H}}^{+}) \cdot \mathbb{P}_{(\widetilde{X_{\mathrm{H}}^{-}}, \widetilde{Y})}\big((\widetilde{X_{\mathrm{H}}^{-}} - \theta)\widetilde{Y} > 0\big) + \mathbb{P}(\widetilde{Y} = -, Y = -|X_{\mathrm{H}}^{-}) \cdot \mathbb{P}_{(\widetilde{X_{\mathrm{H}}^{-}}, \widetilde{Y})}\big((\widetilde{X_{\mathrm{H}}^{-}} - \theta)\widetilde{Y} > 0\big) \\
&= p \cdot \rho_{\mathrm{H}}^{+} \cdot \mathbb{P}_{(X_{\mathrm{H}}^{+}, Y=+)}\big((X_{\mathrm{H}}^{+} - \theta)Y < 0\big) + (1 - p) \cdot (1 - \rho_{\mathrm{H}}^{-}) \cdot \mathbb{P}_{(X_{\mathrm{H}}^{-}, Y=-)}\big((X_{\mathrm{H}}^{-} - \theta)Y > 0\big) \\
&= p \cdot \rho_{\mathrm{H}}^{+} \cdot \mathbb{P}_{(X_{\mathrm{H}}^{+}, Y=+)}\big(X_{\mathrm{H}}^{+} < \theta\big) + (1 - p) \cdot (1 - \rho_{\mathrm{H}}^{-}) \cdot \mathbb{P}_{(X_{\mathrm{H}}^{-}, Y=-)}\big(X_{\mathrm{H}}^{-} < \theta\big) \\
&= p \cdot \rho_{\mathrm{H}}^{+} \cdot \mathrm{Err}_{X_{\mathrm{H}}^{+}} + (1 - p) \cdot (1 - \rho_{\mathrm{H}}^{-}) \cdot (1 - \mathrm{Err}_{X_{\mathrm{H}}^{-}}).
\end{aligned}
$$

$$
\mathrm{Err}_{\widetilde{X_{\mathrm{T}}^{+}}}(f) = p \cdot (1 - \rho_{\mathrm{T}}^{+}) \cdot \mathrm{Err}_{X_{\mathrm{T}}^{+}} + (1 - p) \cdot \rho_{\mathrm{T}}^{-} \cdot (1 - \mathrm{Err}_{X_{\mathrm{T}}^{-}}).
$$

$$
\mathrm{Err}_{\widetilde{X_{\mathrm{T}}^{-}}}(f) = p \cdot \rho_{\mathrm{T}}^{+} \cdot \mathrm{Err}_{X_{\mathrm{T}}^{+}} + (1 - p) \cdot (1 - \rho_{\mathrm{T}}^{-}) \cdot (1 - \mathrm{Err}_{X_{\mathrm{T}}^{-}}).
$$

$\square$

## B.3 PROOF OF THEOREM A.4

*Proof.* For balanced clean prior ($p = 0.5$), we have:

$$
\mathrm{Err}(f) := (1 - \Phi(-\eta)) \cdot \Big( \mathrm{Err}_{X_{\mathrm{H}}^{+}}(f) + \mathrm{Err}_{X_{\mathrm{H}}^{-}}(f) \Big) + \Phi(-\eta) \cdot \Big( \mathrm{Err}_{X_{\mathrm{T}}^{+}}(f) + \mathrm{Err}_{X_{\mathrm{T}}^{-}}(f) \Big),
$$

and

$$
\begin{aligned}
\mathrm{RM:} \quad &\min_{f} \quad \mathbb{P}(\widetilde{X_{\mathrm{H}}^{+}}) \cdot \mathrm{Err}_{\widetilde{X_{\mathrm{H}}^{+}}}(f) + \mathbb{P}(\widetilde{X_{\mathrm{H}}^{-}}) \cdot \mathrm{Err}_{\widetilde{X_{\mathrm{H}}^{-}}}(f) + \mathbb{P}(\widetilde{X_{\mathrm{T}}^{+}}) \cdot \mathrm{Err}_{\widetilde{X_{\mathrm{T}}^{+}}}(f) + \mathbb{P}(\widetilde{X_{\mathrm{T}}^{-}}) \cdot \mathrm{Err}_{\widetilde{X_{\mathrm{T}}^{-}}}(f) \\
&= \min_{f} \quad (1 - \Phi(-\eta)) \cdot \Big( (1 - \rho_{\mathrm{H}}^{+} + \rho_{\mathrm{H}}^{-}) \cdot \mathrm{Err}_{\widetilde{X_{\mathrm{H}}^{+}}}(f) + (1 + \rho_{\mathrm{H}}^{+} - \rho_{\mathrm{H}}^{-}) \cdot \mathrm{Err}_{\widetilde{X_{\mathrm{H}}^{-}}}(f) \Big) \\
&\quad + \Phi(-\eta) \cdot \Big( (1 - \rho_{\mathrm{T}}^{+} + \rho_{\mathrm{T}}^{-}) \cdot \mathrm{Err}_{\widetilde{X_{\mathrm{T}}^{+}}}(f) + (1 - \rho_{\mathrm{T}}^{-} + \rho_{\mathrm{T}}^{+}) \cdot \mathrm{Err}_{\widetilde{X_{\mathrm{T}}^{-}}}(f) \Big).
\end{aligned}
$$

Define the noise rate gaps $\rho_H := \rho_H^+ - \rho_H^-$, $\rho_T := \rho_T^+ - \rho_T^-$, and imbalance ratio $r := \frac{1-\Phi(-\eta)}{\Phi(-\eta)}$, we then have:

RM

$$\iff \min_f \quad r \cdot \left((1-\rho_H) \cdot \mathrm{Err}_{\widetilde{X_H^+}}(f) + (1+\rho_H) \cdot \mathrm{Err}_{\widetilde{X_H^-}}(f)\right) + \left((1-\rho_T) \cdot \mathrm{Err}_{\widetilde{X_T^+}}(f) + (1+\rho_T) \cdot \mathrm{Err}_{\widetilde{X_T^-}}(f)\right)$$

$$\iff \min_f \quad r \cdot \left((1-\rho_H) \cdot \left[(1-\rho_H^+) \cdot \mathrm{Err}_{X_H^+} + \rho_H^- \cdot (1-\mathrm{Err}_{X_H^-})\right] + (1+\rho_H) \cdot \left[\rho_H^+ \cdot \mathrm{Err}_{X_H^+} + (1-\rho_H^-) \cdot (1-\mathrm{Err}_{X_H^-})\right]\right)$$
$$+ \left((1-\rho_T) \cdot \left[(1-\rho_T^+) \cdot \mathrm{Err}_{X_T^+} + \rho_T^- \cdot (1-\mathrm{Err}_{X_T^-})\right] + (1+\rho_T) \cdot \left[\rho_T^+ \cdot \mathrm{Err}_{X_T^+} + (1-\rho_T^-) \cdot (1-\mathrm{Err}_{X_T^-})\right]\right)$$

$$\iff \min_f \quad r \cdot \left(\left[(1-\rho_H^+) \cdot \mathrm{Err}_{X_H^+} + \rho_H^- \cdot (1-\mathrm{Err}_{X_H^-})\right] + \left[\rho_H^+ \cdot \mathrm{Err}_{X_H^+} + (1-\rho_H^-) \cdot (1-\mathrm{Err}_{X_H^-})\right]\right)$$
$$+ \left(\left[(1-\rho_T^+) \cdot \mathrm{Err}_{X_T^+} + \rho_T^- \cdot (1-\mathrm{Err}_{X_T^-})\right] + \left[\rho_T^+ \cdot \mathrm{Err}_{X_T^+} + (1-\rho_T^-) \cdot (1-\mathrm{Err}_{X_T^-})\right]\right)$$
$$- r \cdot \rho_H \cdot \left(\left[(1-\rho_H^+) \cdot \mathrm{Err}_{X_H^+} + \rho_H^- \cdot (1-\mathrm{Err}_{X_H^-}) - \rho_H^+ \cdot \mathrm{Err}_{X_H^+} - (1-\rho_H^-) \cdot (1-\mathrm{Err}_{X_H^-})\right]\right)$$
$$- \rho_T \cdot \left[(1-\rho_T^+) \cdot \mathrm{Err}_{X_T^+} + \rho_T^- \cdot (1-\mathrm{Err}_{X_T^-}) - \rho_T^+ \cdot \mathrm{Err}_{X_T^+} - (1-\rho_T^-) \cdot (1-\mathrm{Err}_{X_T^-})\right]$$

$$\iff \min_f \quad r \cdot \left(\mathrm{Err}_{X_H^+} + \mathrm{Err}_{X_H^-}\right) + \left(\mathrm{Err}_{X_T^+} + \mathrm{Err}_{X_T^-}\right)$$
$$- r \cdot \rho_H \cdot \left(\left[(1-2\rho_H^+) \cdot \mathrm{Err}_{X_H^+} - (1-2\rho_H^-) \cdot (1-\mathrm{Err}_{X_H^-})\right]\right)$$
$$- \rho_T \cdot \left[(1-2\rho_T^+) \cdot \mathrm{Err}_{X_T^+} - (1-2\rho_T^-) \cdot (1-\mathrm{Err}_{X_T^-})\right]$$

$$\iff \min_f \quad \mathrm{Err}(f) - r \cdot \rho_H \cdot \left(\left[(1-2\rho_H^+) \cdot \mathrm{Err}_{X_H^+} - (1-2\rho_H^-) \cdot (1-\mathrm{Err}_{X_H^-})\right]\right)$$
$$- \rho_T \cdot \left[(1-2\rho_T^+) \cdot \mathrm{Err}_{X_T^+} - (1-2\rho_T^-) \cdot (1-\mathrm{Err}_{X_T^-})\right]. \tag{4}$$

To achieve relatively fair performances between $\widetilde{X_T^+}$ and $\widetilde{X_T^-}$, i.e., the performance gap between $\mathrm{Err}_{\widetilde{X_T^+}}(f)$ and $\mathrm{Err}_{\widetilde{X_T^-}}(f)$ is supposed to be small. Note that:

$$\mathrm{Err}_{\widetilde{X_T^+}}(f) - \mathrm{Err}_{\widetilde{X_T^-}}(f)$$
$$= \left[p \cdot (1-\rho_T^+) \cdot \mathrm{Err}_{X_T^+} + (1-p) \cdot \rho_T^- \cdot (1-\mathrm{Err}_{X_T^-})\right] - \left[p \cdot \rho_T^+ \cdot \mathrm{Err}_{X_T^+} + (1-p) \cdot (1-\rho_T^-) \cdot (1-\mathrm{Err}_{X_T^-})\right]$$
$$= \frac{1}{2} \cdot \left[(1-2\rho_T^+) \cdot \mathrm{Err}_{X_T^+} - (1-2\rho_T^-) \cdot (1-\mathrm{Err}_{X_T^-})\right].$$

Similarly, for the two head sub-populations, we could derive that:

$$\mathrm{Err}_{\widetilde{X_H^+}}(f) - \mathrm{Err}_{\widetilde{X_H^-}}(f)$$
$$= \left[p \cdot (1-\rho_H^+) \cdot \mathrm{Err}_{X_H^+} + (1-p) \cdot \rho_T^- \cdot (1-\mathrm{Err}_{X_H^-})\right] - \left[p \cdot \rho_H^+ \cdot \mathrm{Err}_{X_H^+} + (1-p) \cdot (1-\rho_H^-) \cdot (1-\mathrm{Err}_{X_H^-})\right]$$
$$= \frac{1}{2} \cdot \left[(1-2\rho_H^+) \cdot \mathrm{Err}_{X_H^+} - (1-2\rho_H^-) \cdot (1-\mathrm{Err}_{X_H^-})\right].$$

Thus, by incorporating the above two performance gaps into Eqn. (4), the RM has its equivalent form as below:

$$\mathrm{RM} \iff \min_f \quad \mathrm{Err}(f) - r \cdot \rho_H \cdot \left(\left[(1-2\rho_H^+) \cdot \mathrm{Err}_{X_H^+} - (1-2\rho_H^-) \cdot (1-\mathrm{Err}_{X_H^-})\right]\right)$$
$$- \rho_T \cdot \left[(1-2\rho_T^+) \cdot \mathrm{Err}_{X_T^+} - (1-2\rho_T^-) \cdot (1-\mathrm{Err}_{X_T^-})\right]$$
$$\iff \min_f \quad \mathrm{Err}(f) - 2r \cdot \rho_H \cdot \left(\mathrm{Err}_{\widetilde{X_H^+}}(f) - \mathrm{Err}_{\widetilde{X_H^-}}(f)\right) - \rho_T \cdot \left(\mathrm{Err}_{\widetilde{X_T^+}}(f) - \mathrm{Err}_{\widetilde{X_T^-}}(f)\right).$$

$$\square$$

## C  ADDITIONAL EXPERIMENT RESULTS AND DETAILS

### C.1  INFLUENCES OF SUB-POPULATIONS ON TEST DATA

**Influences on Class-Level Test Accuracy**  When removing all samples from the sub-population $\mathcal{G}_i$ during the whole training procedure, remember that we defined the (class-level) test accuracy changes as:

$$\text{Acc}_\text{c}(\mathcal{A}, \widetilde{S}, i, j) := \mathbb{P}_{\substack{f \leftarrow \mathcal{A}(\widetilde{S}) \\ (X', Y'=j)}} (f(X') = Y') - \mathbb{P}_{\substack{f \leftarrow \mathcal{A}(\widetilde{S}^{\backslash i}) \\ (X', Y'=j)}} (f(X') = Y'),$$

where $(X', Y' = j)$ denotes the test data distribution given the clean label $j$. In Figure 8, the $x$-axis denotes the loss function for training, and the $y$-axis visualized the distribution of $\{\text{Acc}_\text{c}(\mathcal{A}, S, i, j)\}_{j \in [10]}$ (above the dashed line) and $\{\text{Acc}_\text{c}(\mathcal{A}, \widetilde{S}, i, j)\}_{j \in [10]}$ (below the dashed line) for several long-tailed sub-populations ($i = 52, 37, 75, 19, 81, 36, 91, 63, 70, 55, 67, 41, 98, 40, 61, 87, 71$) under each robust method. The blue zone shows the 25-th percentile ($Q_1$) and 75-th percentile ($Q_3$) accuracy changes, and the orange line indicates the median value. Accuracy changes that are drawn as circles are viewed as outliers. Note that all sub-figures have the same limits for $y$-axis, Observation 3.1 holds for more tail sub-populations under class-level test accuracy as well.

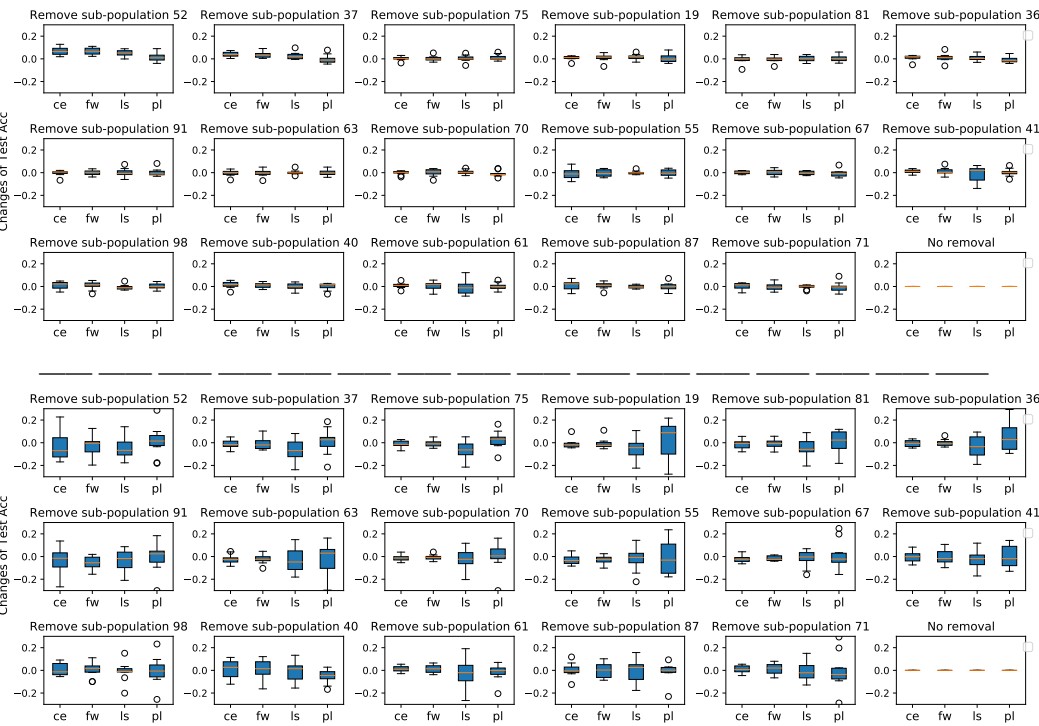

Figure 8: (Completed version) Box plot of the class-level test accuracy changes when removing all samples of a selected long-tailed sub-population during the training w.r.t. 4 methods. (Above the dashed line: trained on clean labels; below the dashed line: trained on noisy labels.)

**Influences on Population-Level Test Accuracy**  When removed all samples from the sub-population $\mathcal{G}^{(i)}$ during the whole training procedure, remember the (population-level) test accuracy changes is defined as:

$$\text{Acc}_\text{p}(\mathcal{A}, \widetilde{S}, i, j) := \mathbb{P}_{\substack{f \leftarrow \mathcal{A}(\widetilde{S}) \\ (X', Y', G=j)}} (f(X') = Y') - \mathbb{P}_{\substack{f \leftarrow \mathcal{A}(\widetilde{S}^{\backslash i}) \\ (X', Y', G=j)}} (f(X') = Y'),$$

where $(X', Y', G = j)$ indicates the test data distribution given that the samples are from the $j$-th population. In Figure 9, we repeat the previous step while visualize the distribution of

$\{\text{Acc}_p(\mathcal{A}, S, i, j)\}_{j \in [100]}$ (Above the dashed line) and $\{\text{Acc}_p(\mathcal{A}, \widetilde{S}, i, j)\}_{j \in [100]}$ (Below the dashed line). Similarly, by referring to the wide range of box plotted distributions, Observation 3.1 holds for more tail sub-populations as well. Besides, the variance and the extremer of the changes in the test accuracy are much larger in the view of sub-populations.

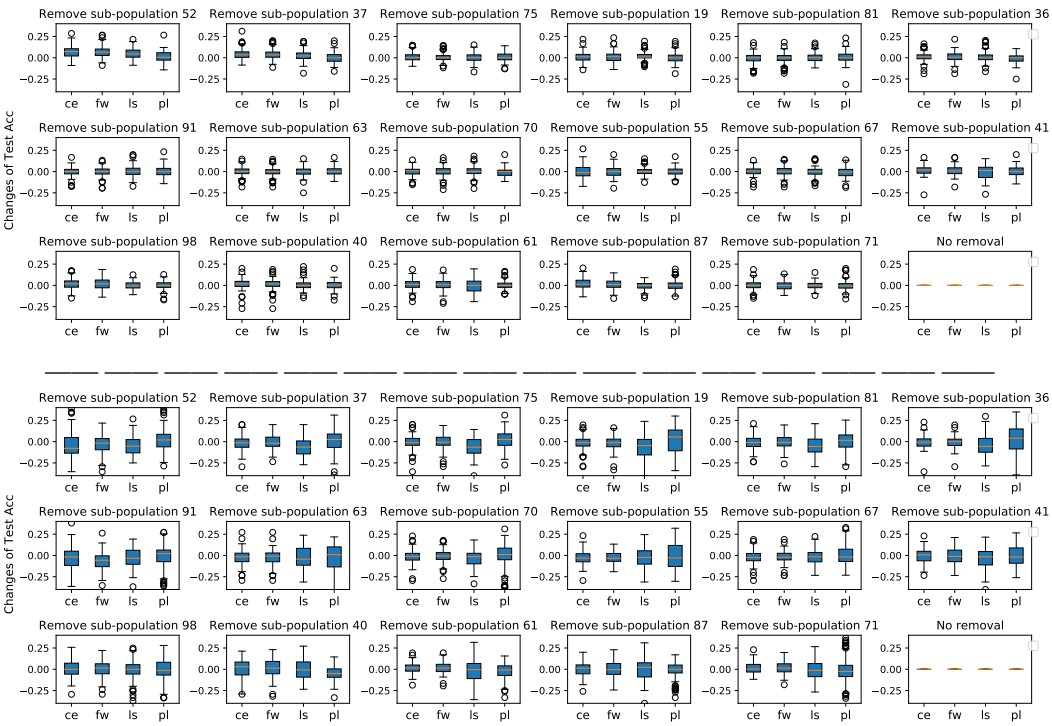

Figure 9: (Complete version) Box plot of the population-level test accuracy changes when removing all samples of a selected long-tailed sub-population during the training w.r.t. 4 methods. (Above the dashed line: trained on clean labels; Below the dashed line: trained on noisy labels.)

**Influences on Sample-Level Prediction Confidence** Remember that we characterize the influence of a sub-population on a test sample as:

$$\text{Infl}(\mathcal{A}, \widetilde{S}, i, j) := \mathbb{P}_{f \leftarrow \mathcal{A}(\widetilde{S})}(f(x'_j) = y'_j) - \mathbb{P}_{f \leftarrow \mathcal{A}(\widetilde{S}^{\setminus i})}(f(x'_j) = y'_j),$$

where in the above two quantities, $f \leftarrow \mathcal{A}(\widetilde{S})$ denotes that the classifier $f$ is trained from the whole noisy training dataset $\widetilde{S}$ via Algorithm $\mathcal{A}$, $f \leftarrow \mathcal{A}(\widetilde{S}^{\setminus i})$ means $f$ got trained on $\widetilde{S}$ without samples in the sub-population $\mathcal{G}^{(i)}$. And $\text{Infl}(\mathcal{A}, \widetilde{S}, i, j)$ quantifies the influence of a certain sub-population on a specific test data. As shown in Figure 10, we visualize $\text{Infl}(\mathcal{A}, S, i, j)$ (1st row) and $\text{Infl}(\mathcal{A}, \widetilde{S}, i, j)$ (2nd row), where $j \in [10000]$ means 10K test samples. With the presence of label noise, Observations 3.2 holds for more tail sub-population as well.

## C.2 EXPERIMENT DETAILS ON CIFAR DATASETS

The original CIFAR-10 (Krizhevsky et al., 2009) dataset contains 60k $32 \times 32$ color images, 50k images for training, and 10k images for testing. The dataset is balanced and each image belongs to one of ten completely mutually exclusive classes. CIFAR-100 dataset shares the same statistics, except for containing 100 completely mutually exclusive classes.

**Hyper-Parameter Settings** For each baseline method, we adopt mini-batch size 128, optimizer SGD, initial learning rate 0.1, momentum 0.9, weight decay 0.0005, number of epochs 200. As for the learning rate scheduler, we followed (Cui et al., 2019) and chose a linear warm-up of learning rate

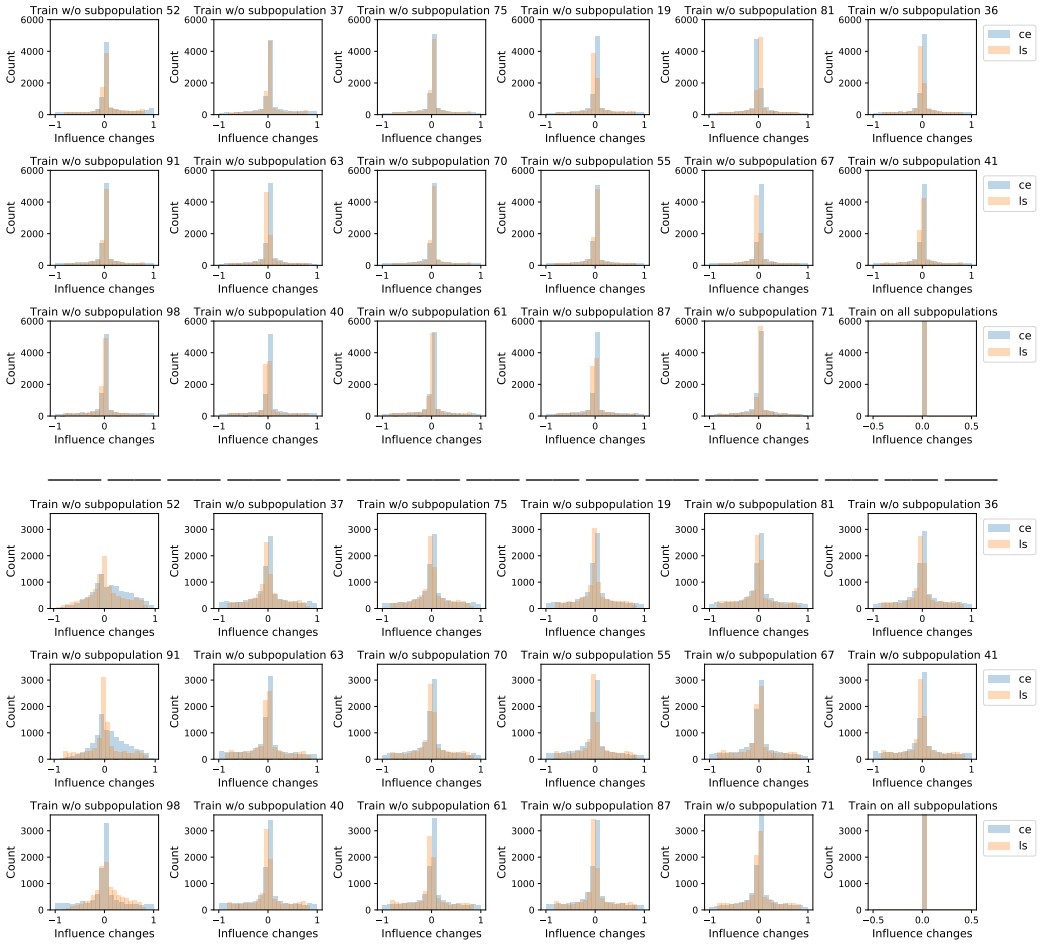

Figure 10: (Complete version) Distribution plot w.r.t. the changes of model confidence on the test data samples using CE loss and label smoothing (Above the dashed line: trained on clean labels; Below the dashed line: trained on noisy labels).

(Goyal et al., 2017) in the first 5 epochs, then decay 0.01 after the 160-th epoch and 180-th epoch. Standard data augmentation is applied to each synthetic CIFAR dataset. We did not make use of any advanced re-sampling strategies or data augmentation techniques. All experiments run on a cluster of Nvidia RTX A5000 GPUs.

**The Value of $\lambda$ in FR (KNN)** We tuned the performance of FR (KNN) w.r.t. a set $\{0.0, 0.1, 0.2, 0.4, 0.6, 0.8, 1.0, 2.0\}$, where $\lambda = 0.0$ indicates the training of baseline methods without **FR**. Regarding the reported results in the main paper: for all methods w.r.t. CIFAR-100 dataset, we set $\lambda = 0.1$ since calculating the accuracy of tail sub-populations may be unstable. As for methods on CIFAR-10 Imb noise, we set $\lambda = 1.0$ for CE loss, NLS, Focal loss and Peer Loss. One exception is that LS requires a relative small $\lambda$, i.e., $\lambda = 0.4$. Also, we observe that for Imb noise, a larger $\lambda$ could be more beneficial for CE loss and Logit-adj loss under a higher noise regime. For methods on CIFAR-10 Sym noise, the $\lambda$ selection for LS, NLS, PeerLoss remains the same as that in the Imb setting. For CE and Focal loss, a larger $\lambda$ (i.e., $\lambda = 2.0$) could be more beneficial. While Logit-adj prefers a smaller $\lambda$ (i.e., $\lambda = 0.5$).

**The Value of $\lambda$ in FR (G2)** Since there are only two sub-populations considered, the experiment results on CIFAR-100 would be more stable than FR (KNN), hence a larger $\lambda$ could be utilized. For CE loss, NLS and Focal loss, we adopt $\lambda = 1.0$ for all CIFAR-10 experiments and $\lambda = 2.0$ for all CIFAR-100 experiments. For LS, we set $\lambda = 0.4$ for all experiments, except for the extreme case (CIFAR-100 with large noise $\rho = 0.5$), where we decide on a larger $\lambda$ (0.8). As for PeerLoss, we have to set a relatively small $\lambda$ (i.e., $\lambda = 0.8$) for CIFAR-10 experiments and an even smaller one

($\lambda = 0.2$) on CIFAR-100 due to the scale of its loss. And we set $\lambda = 1.0$ for Logit-adj under all settings. We do observe better results when adopting varied $\lambda$ under each setting, but fixing the $\lambda$ for reporting under a specific dataset tends is more convenient and practical.

**Hypothesis Testing w.r.t. FR** We adopt paired student t-test to verify the conclusion that **FR** helps with improving the learning of the classifier (i.e., test accuracy). Briefly speaking, for each dataset and each baseline method, we statistically test whether **FR** results in significant test accuracy improvements in Table 1.

Denote by $\text{PA}_{\text{Method}}^{\rho,r} := (\text{Acc}_{\text{Method}}^{\rho,r}, \text{Acc}_{\text{Method+FR}}^{\rho,r})$ the Paired Accuracies without/with **FR** under each setting, i.e., when $\rho = 0.2, r = 10$, Method=CE, we have: $\text{PA}_{\text{Method}}^{\rho,r} = (79.75, 80.46)$ (w.r.t. FR (KNN)). The null and alternative hypotheses could be expressed as:

$$\mathbf{H_0} : \{\text{Acc}_{\text{Method+FR}}^{\rho,r}\}_{\rho,r} \text{ come from the } \textit{same} \text{ distribution as } \{\text{Acc}_{\text{Method}}^{\rho,r}\}_{\rho,r};$$

$$\mathbf{H_1} : \{\text{Acc}_{\text{Method+FR}}^{\rho,r}\}_{\rho,r} \text{ come from } \textit{different} \text{ distributions as } \{\text{Acc}_{\text{Method}}^{\rho,r}\}_{\rho,r},$$

where the above accuracy list $\{\text{Acc}_{\text{Method}}^{\rho,r}\}_{\rho,r}$ includes both noise types (imb & sym), $\rho \in [0.2, 0.5]$, and $r \in [10, 50, 100]$, thus 12 elements for either dataset, similarly for the accuracy list $\{\text{Acc}_{\text{Method+FR}}^{\rho,r}\}_{\rho,r}$.

## C.3 EXPERIMENT DETAILS ON CLOTHING1M DATASET

We adopt the same baselines as reported in CIFAR experiments for Clothing1M. All methods use the pre-trained ResNet50, optimizer SGD, momentum 0.9, and weight decay 1e-3. The initial learning rate is 0.01, then it decays 0.1 for every 30 epochs so there are 120 epochs in all. Negative Label Smoothing (NLS) (Wei et al., 2021b) resumes the last epoch training of CE, and proceeds to train with NLS for another 40 epochs (learning rate 1e-7).

## C.4 INFLUENCES OF HEAD SUB-POPULATIONS ON TEST DATA

Table 4 briefly introduces the influences of head populations ($> 500$ samples) on the overall test accuracy. Clearly, the mentioned 5 head populations have different impacts: with the presence of label noise, Pop-06 becomes harmful while Pop-02 and Pop-04 remain helpful.

Table 4: Influences of head populations on the overall test accuracy.

| Clean | Remove Pop-02 | Remove Pop-04 | Remove Pop-03 | Remove Pop-06 | Remove Pop-00 |
|---|---|---|---|---|---|
| Test Acc | -4.06 | -3.09 | -0.96 | -0.15 | +0.29 |

| Noisy $\rho = 0.2$ | Remove Pop-02 | Remove Pop-04 | Remove Pop-03 | Remove Pop-06 | Remove Pop-00 |
|---|---|---|---|---|---|
| Test Acc | -3.75 | -4.80 | -3.79 | +0.62 | +0.18 |

## C.5 EXPERIMENTS ON LONG-TAILED DATA WITH REAL-WORLD NOISY LABELS

We further provide more experiment results on real-world noisily labeled long-tailed data, including long-tailed CIFAR-10N, CIFAR-20N, CIFAR-100N, and Animal-10N.

**Dataset Statistics** Denote by $\rho$ the percentage of wrong labels among the training set, CIFAR-10N (Wei et al., 2022c) provides three types of real-world noisy human annotations on the CIFAR-10 training dataset, with $\rho = 0.09, 0.18, 0.40$. CIFAR-100 N (Wei et al., 2022c) provides each CIFAR-100 training image with a human annotation where $\rho = 0.40$. In CIFAR-20N (Wei et al., 2022c) ($\rho = 0.25$), each training/test image includes a coarse label (among 20 coarse classes) and a finer label (among 100 fine classes). We view the coarse label as the class information for training, and we illustrate the effectiveness of all methods by referring to their averaged performance on 100 groups at the test time. Neither the number of groups nor the group information of each image is utilized during the training. Animal-10N (Song et al., 2019) dataset is a ten classes classification task including 5 pairs of confusing animals with a total of 55,000 images. The simulation of long-tailed samples follows the same procedure as the results in Table 1.

Table 5: Performance comparisons on long-tailed datasets with real-world noisy labels, best-achieved test accuracy are reported. Results in **bold** mean **FR** improves the performance of the baseline methods, respectively.

| | CIFAR-10N (Agg) | CIFAR-10N (Rand1) | CIFAR-10N (Worse) | CIFAR-100N | CIFAR-20N | Animal-10N |
|---|---|---|---|---|---|---|
| **Noise type: Real-World Human Noise** | | | | | | |
| Imbalance Ratio ($r = 10$) | CIFAR-10N (Agg) | CIFAR-10N (Rand1) | CIFAR-10N (Worse) | CIFAR-100N | CIFAR-20N | Animal-10N |
| CE | 83.60 | 81.53 | 71.83 | 43.10 | 60.08 | 71.18 |
| CE + FR (G2) | **83.69** | **81.86** | **74.67** | **44.79** | **61.24** | **72.04** |
| Logit-adj (Menon et al., 2021) | 84.03 | 78.85 | 64.41 | 41.93 | 58.98 | 67.78 |
| Logit-adj + FR (G2) | **84.88** | **80.37** | **65.15** | **43.48** | **59.89** | **69.64** |
| Imbalance Ratio ($r = 50$) | CIFAR-10N (Agg) | CIFAR-10N (Rand1) | CIFAR-10N (Worse) | CIFAR-100N | CIFAR-20N | Animal-10N |
| CE | 74.54 | 71.55 | 61.36 | 32.51 | 50.83 | 52.60 |
| CE + FR (G2) | **74.93** | **73.22** | **65.01** | **34.49** | **51.93** | 51.88 |
| Logit-adj (Menon et al., 2021) | 75.67 | 66.15 | 52.93 | 30.06 | 46.83 | 56.28 |
| Logit-adj + FR (G2) | **75.81** | 65.94 | **54.36** | **30.47** | **47.00** | **62.18** |
| Imbalance Ratio ($r = 100$) | CIFAR-10N (Agg) | CIFAR-10N (Rand1) | CIFAR-10N (Worse) | CIFAR-100N | CIFAR-20N | Animal-10N |
| CE | 69.09 | 64.58 | 57.40 | 29.04 | 45.64 | 42.64 |
| CE + FR (G2) | **69.75** | **66.43** | **59.14** | **31.97** | **46.70** | **45.28** |
| Logit-adj (Menon et al., 2021) | 71.35 | 61.96 | 47.86 | 26.91 | 40.80 | 13.06 |
| Logit-adj + FR (G2) | 70.39 | 59.19 | 47.48 | **28.12** | **41.09** | **48.30** |

**Hyperparameters** The training of CIFAR-10N, CIFAR-20N, and CIFAR-100N is the same as that of synthetic noisy CIFAR datasets. For Animal10N, we adopt VGG19, a different backbone from ResNet. In Animal10N, the settings follow the work (Song et al., 2019): we use VGG-19 with batch normalization and the SGD optimizer. The network trained 100 epochs and we adopted an initial learning rate of 0.1, which is divided by 5 at 50% and 75% of the total number of epochs.

**Results** As shown in the three tables below, we report the test accuracy on the class-balanced dataset with clean labels, as we have done in Table 1 of the main paper. We adopt $\lambda = 2$ for CE+FR for all settings and $\lambda = 1$ for Logit-adj loss for all settings. Experiment results show that FR helps with improving the test accuracy in most settings, given CE loss or logit-adj loss. And we can conclude that constraining the classifier to have relative fairness performances is beneficial when learning with noisy and long-tailed data.

## D    COMPARISONS WITH MORE RECENT WORKS

We attach detailed discussion about several most-recent related works about learning from long-tailed data with noisy labels as below.

The literature has observed recent efforts in learning with long-tailed data with noisy labels, for example, Wei et al. (2021c) proposed a new prototypical noise detection method which robustly detects label noise in long-tailed learning data, leveraging semi-supervised learning algorithms to further improve generalization. However, the coupling effect of label noise and class imbalance, which is a central theme in our work, is not explicitly considered in their methodology. Later Karthik et al. (2021) bring the insight of self-supervised learning to cope with class imbalance and label noise. Prototypical Classifier is another recent approach (Wei et al., 2022e) which is designed to produce balanced predictions for all classes and detects noisy labels by thresholding the predicted scores of examples. Despite its benefits, it fails to provide a universal improvement across different sub-populations when label noise is present. The threshold adjustment technique is useful but does not account for discrepancies in performance improvement across different sub-populations. More recently, RCAL (Zhang et al., 2022b) method calibrates representation by assuming that the representations of instances in each class conform to a Gaussian distribution. By recovering underlying representation distributions from mislabeled and class-imbalanced data, they improve classifier performance. Still, the method is limited by the assumption of the Gaussian distribution and does not explicitly consider the impact of the interplay between class imbalance and label noise on different sub-populations.

