# OpenReview forum: "Fairness Improves Learning from Noisily Labeled Long-Tailed Data"
_ICLR.cc/2024/Conference — Submitted to ICLR 2024_

### Official Review · Reviewer_Ed4g · 2023-10-31

**Soundness:** 2 fair
**Presentation:** 2 fair
**Contribution:** 2 fair
**Rating:** 3
**Confidence:** 4

**Summary:**

The paper assesses whether fairness methods can improve learning on noisy imbalanced data. A regularization approach is taken to promote similar losses across sub-populations.

**Strengths:**

- The claim is interesting and well-motivated: that fairness improves learning on noisy long-tail sub-populations.
- Extensive experiments are conducted.

**Weaknesses:**

- Table 3: empirical improvements seem marginal and lack confidence intervals or pairwise statistical significance tests.
  - I'm not too convinced that, e.g., changes from 74.46 to 74.48-74.50 are significant;
  - I suggest bolding only cells whose difference to the baseline are statistical significant, as opposed to a simple "larger than" test.
  - As an example, you can get confidence intervals using bootstrapping on the test set, without retraining; or running k-fold cross-validation to obtain a measure of dispersion of results.

- It's not clear whether access to group membership is useful given that the work is not targeting fairness improvements.
  To target the noisy long-tail sub-populations couldn't we just up-weight samples with higher loss and down-weight samples
  with lower loss? (an approach that is also quite common in the fairness literature)
  - What is the intuition behind the use of group membership supposedly leading to better performance?

- Fig. 6: there seems to be a lot of noise on the effect of FR across sub-populations;
  the paper is missing a more in depth analysis of the mechanism by which overall performance is increased.
  - Intuitively, it should improve performance on populations with lower representation in the dataset (tail populations);
    one example plot would show performance change on the y-axis (acc. increase or decrease after introducing FR), and sub-population size on the x-axis.

- Using a regularizer on the performance differences among sub-populations is not particularly novel.
  - The novelty stems from the use of a standard fairness tool to improve learning performance
    (regardless of fairness); but the fact that this is a standard fairness tool (and that FR is not
    the focus of the paper in any way) should really be made more clear in the paper.
  - Perhaps, if the goal is to show that standard fairness tools can have a positive impact on "noisy
    long-tailed sub-populations", it'd make more sense to use unaltered fair ML methods (that aim to
    bring sub-population performance closer) and show that the paper claim on improved learning holds.
  - Essentially, most fair ML methods reduce to the same thing as FR: up-weighing samples whose loss
    is larger than the average loss, or samples whose group has a larger loss than average.

**Questions:**

- Fig. 3: does the index have any meaning? Should this be sorted by frequency?
- Fig. 4: please use a more adequate font size, it's completely unreadable when printed.
- Instead of holding all $\lambda_i$ constant it would be interesting to use a standard dual ascent framework to jointly optimize the Lagrange multipliers.
  - "Intuitively, applying dual ascent is likely to result in a large $\lambda_i$ on the worst sub-population, inducing possible negative effects"
  - It'd be interesting to back this up with actual results.
- Eq. 3: given that the constraint that is actually used in practice is different from the definition of $\text{Dist}_i$ given in Section 4.1., I'd suggest giving a different name to each of these constraints.

---

> ### Author Response · Authors · 2023-11-21
> **Response to Reviewer Ed4g (Part 1)**
>
> Thanks for your time for reviewing our work! Please refer to our following responses regarding your insightful suggestions!
>
> ```Weaknesses 1:```
>
> [Table 3: empirical improvements seem marginal and lack confidence intervals or pairwise statistical significance tests.]
>
> ```Response:```
>
> **Why there is no statistical test in Table 3**
>
> Our main results are summarized in Table 1, whose statistical tests are given in Table 2. Only Table 3 is not provided with hypothesis testing, since our aim for giving Table 3 is the ablation study – we want to show $\lambda>0$ generally improves learning from the noisily labeled long-tailed data. What is more, Clothing 1M is not a standard benchmark like MNIST/CIFAR and it has a single realization.
>
> **New highlight criterion**
>
> We recognize the need for a more robust method to highlight significant improvements. In light of your suggestion, we plan to revise our highlighting criteria. In the revised manuscript, we will only highlight cells in Table 3 where the increase in test accuracy is greater than 0.1. We believe this threshold represents a substantial difference, particularly in large-scale datasets such as ImageNet and Clothing-1M. This change aims to provide a clearer and more meaningful distinction of the results that are statistically significant.
>
> ```Weaknesses 2:```
>
> [It's not clear whether access to group membership is useful given that the work is not targeting fairness improvements. To target the noisy long-tail sub-populations couldn't we just up-weight samples with higher loss and down-weight samples with lower loss? (an approach that is also quite common in the fairness literature). What is the intuition behind the use of group membership supposedly leading to better performance?]
>
> ```Response:```
>
> **Addressing Sample-wise Reweighting** In our previous research, we have observed that in scenarios involving long-tailed data with label noise, samples with small losses often correlate with major populations and clean labels. Conversely, samples with large losses could either belong to minor populations or be incorrectly labeled. Therefore, simply up-weighting high-loss samples could inadvertently amplify the influence of mislabeled data, leading to skewed model predictions. This insight influenced our decision to not solely rely on sample-wise reweighting.
>
> **Explaining the Absence of Class-level Consideration**
>
> We also steered clear of class-level reweighting, primarily due to the noisy nature of class labels in our dataset. Given this uncertainty in labeling, class-level adjustments could potentially introduce additional biases.
>
> **Justifying Sub-population Level Approach**
>
> Our focus on sub-population levels stems from their likelihood to be feature-dependent, circumventing the need to evaluate class or sample-wise performance directly. We aimed to illustrate this concept through Figure 1, which demonstrates how head sub-populations generally benefit from existing methods, while tail sub-populations could see improvements with the implementation of a fairness regularizer.
>
> **Clarifying the Novelty of Our Approach**
>
>  Unlike the traditional fairness-accuracy tradeoff in resource-constrained settings, our findings, as presented in Figure 2 of the main paper, suggest that current methodologies for handling long-tailed noisy data fail to achieve Pareto optimality. This results in significant performance discrepancies across different classes/populations, hinting at an unfair distribution of model performance. We hypothesize that robust solutions might disproportionately affect sub-populations. Section 3 of our paper delves into this hypothesis by examining the influence of sub-populations on test data, revealing distinct impacts on model performance when dealing with noisily labeled long-tail data.
>
> **Presenting the Fairness Regularizer**
>
> Motivated by these observations, Section 4 introduces our Fairness Regularizer (FR). This novel approach is designed to prevent overemphasis on specific sub-populations and to minimize performance gaps across different groups. This is particularly crucial in the context of long-tailed, noisily labeled data, where certain populations might be significantly overlooked without fairness constraints. Our empirical studies (Section 5) and theoretical analysis (Appendix A) support the assertion that implementing FR can enhance the performance of tail sub-populations with minimal impact on head sub-populations.
>
> **Theoretical Underpinnings in Appendix A**
>
> In Appendix A, we provide a theoretical exploration of the effects of sub-populations in learning with long-tailed noisy data, using a binary Gaussian example. Our theorem suggests that by enforcing fair performance across specific sub-populations (i.e., equal training error probabilities), a classifier trained on noisy data with a fairness regularizer can approximate the optimal classifier for the clean data distribution.

---

> > ### Author Response · Authors · 2023-11-21
> > **Response to Reviewer Ed4g (Part 2)**
> >
> > ```Weaknesses 3:```
> >
> > [Fig. 6: there seems to be a lot of noise on the effect of FR across sub-populations; the paper is missing a more in depth analysis of the mechanism by which overall performance is increased. Intuitively, it should improve performance on populations with lower representation in the dataset (tail populations); one example plot would show performance change on the y-axis (acc. increase or decrease after introducing FR), and sub-population size on the x-axis.]
> >
> > ```Response:```
> >
> > Thanks for the great suggestion. In our revision, we will also include the sub-population level performance changes for a more clear comparison, as we have done in Figure 2.
> >
> > ```Weaknesses 4:```
> >
> > [Using a regularizer on the performance differences among sub-populations is not particularly novel. The novelty stems from the use of a standard fairness tool to improve learning performance (regardless of fairness); but the fact that this is a standard fairness tool (and that FR is not the focus of the paper in any way) should really be made more clear in the paper. Perhaps, if the goal is to show that standard fairness tools can have a positive impact on "noisy long-tailed sub-populations", it'd make more sense to use unaltered fair ML methods (that aim to bring sub-population performance closer) and show that the paper claim on improved learning holds. Essentially, most fair ML methods reduce to the same thing as FR: up-weighing samples whose loss is larger than the average loss, or samples whose group has a larger loss than average.]
> >
> > ```Response:```
> >
> >
> > **Clarification about the novelty**
> >
> >
> > We firstly want to clarify our main contributions/novelty as below:
> >
> >
> > * We explore the setting of learning with long-tailed noisily labeled data. Note that existing classification datasets are often solicited from the imperfect human annotation or pseudo-machine-generated labels, unfortunately, most works focus on the class-balanced setting. Our work considers the setting where both challenges appear (long-tailed data distribution, noisy label). And we show the severe population-level unfair model behavior when both challenges appear (Section 3).
> >
> >
> > * Inspired by the unfair model performance, we then explore how fairness constraints mitigate the impact of long-tailed data distribution along with noisy labels. Empirically, we observe that our introduced fairness regularizer is easy to use and complements existing robust/long-tailed solutions. With extensive empirical results to verify (Sections 5 and C.5), including 6 baselines, 5 datasets (CIFAR-10, CIFAR-20, CIFAR-100, Animal-10N, Clothing-1M), various noise-models (2 synthetic models and real-world noise), backbones (VGG19 on Aninal10N and Resnet for the rest).
> >
> >
> > * **Theoretical guarantees:** It is a common notion that when learning with clean data, increasing the fairness of attribute performance may lead to a large performance drop on the majority group and the averaged performance, even though the performance of the minority group increases. In Appendix A, we adopt a binary gaussian model to show a **novel result:** when learning with long-tailed data with noisy labels, achieving fairness performance among groups could help with improving the averaged model performance.
> >
> > **Differentiation from Existing Fair ML Methods**
> >
> > As mentioned by **Reviewer MjTZ**, most relevant to us is reduction approaches in fair ML methods. In our revision, we will highlight the main differences between FR and the reductions approach. Briefly speaking, typical reduction approaches for statistical parity focus more on achieving statistical equality across groups, possibly through different mechanisms in the training process. When applied to our paper, the main task of reduction approaches is to dealing with the lagrangian:
> > $$\mathbb{E}\_{(X,  \widetilde{Y})\sim \widetilde{\mathcal{D}}}[\ell(f(X), \widetilde{Y})] +\sum_{i=1}^{N}\lambda_i (\text{Dist}_i-\delta),$$
> >
> > and aims to solve the alternative of the original optimization problem of Eqn. (2).
> >
> > $$\min_{f:  X\rightarrow [K]} \mathbf{E}_{(X, \widetilde{Y})\sim \widetilde{\mathcal{D}}} [\ell(f(X), \widetilde{Y})], \qquad s.t. \qquad \text{Dist}_i\leq \delta, \forall i \in [N], \text{[Eqn. (2)]} $$
> >
> > FR is equivalent to optimizing the lagrangian in Eqn. (2) with $\lambda_i=\lambda$ and $\delta=0$. Although the objects of FR and reduction approaches are close to each other, their optimal solutions differ. This is because our goal is to improve the overall performance on clean and balanced test data, it is arguably a better strategy to not over-addressing the worst sub-population. Hence, we do not optimize for strict fairness among sub-populations as reduction approaches do.
> >
> >
> > [1] Agarwal et al. A Reductions Approach for Fair Classification. 2018.

---

> > > ### Author Response · Authors · 2023-11-21
> > > **Response to Reviewer Ed4g (Part 3)**
> > >
> > > ```Question 1:```
> > >
> > > [Fig. 3: does the index have any meaning? Should this be sorted by frequency?]
> > >
> > > ```Response:```
> > >
> > >
> > > In Figure 3, we perform long-tail selection for 20 coarse classes of CIFAR-20; the smaller class number has many more samples. And we show how many samples are selected from each fine-class (100 classes/sub-populations), x-axis is just the sub-population index. In our revision, we will sort it by frequency for a more clear presentation.
> > >
> > >
> > > ```Question 2:```
> > >
> > > [Fig. 4: please use a more adequate font size, it's completely unreadable when printed.]
> > >
> > > ```Response:```
> > >
> > >
> > > Sorry for the font size. We will use a more adequate font size in our revision.
> > >
> > > ```Question 3:```
> > >
> > > [Instead of holding all $\lambda_i$ constant, it would be interesting to use a standard dual ascent framework to jointly optimize the Lagrange multipliers.
> > > "Intuitively, applying dual ascent is likely to result in a larger $\lambda_i$ on the worst sub-population, inducing possible negative effects". It'd be interesting to back this up with actual results.]
> > >
> > > ```Response:```
> > >
> > > For the first half: [Instead of holding all $\lambda_i$ constant, it would be interesting to use a standard dual ascent framework to jointly optimize the Lagrange multipliers.]
> > >
> > > Thank you for your insightful suggestion regarding the use of a standard dual ascent framework in optimizing the Lagrange multipliers. This perspective indeed offers a compelling direction for further exploration. To address this, let's revisit our original optimization task (Eqn. (2)):
> > >
> > >
> > > $$\min_{f:  X\rightarrow [K]} \mathbf{E}_{(X, \widetilde{Y})\sim \widetilde{\mathcal{D}}} [\ell(f(X), \widetilde{Y})],  \qquad s.t.  \qquad\text{Dist}_i\leq \delta, \forall i \in [N], \qquad \text{[Eqn. (2)]} $$
> > >
> > > An effective approach could involve the augmented Lagrangian method, where we transform the constrained problem into a series of unconstrained minimization problems. Particularly, the penalty method for solving the constrained problem is defined as:
> > >
> > > $$\min_{f: X\to [K]} ~\Phi(f\_{k}):=\mathbb{E}\_{(X, \widetilde{Y})\sim  \widetilde{\mathcal{D}}}[\ell(f\_{k}(X), \widetilde{Y})]+ \mu\_{k} \cdot \sum\_{i\in[N]}g( \text{Dist}\_{i}-\delta), \text{[Eqn. (R1)]}$$
> > >
> > > where $g(x):=\max(0, x)^2$. $k$ indicates the $k$-th iteration while solving a series of unconstrained minimization problems for the original optimization task (Eqn. (2)), in other words, solving the unconstrained problem and use the solution as the initial guess for the next iteration. Solutions of the successive unconstrained problem will asymptotically converge to the original optimization task (Eqn. (2)).
> > >
> > > Setting $\delta=0$, the augmented Lagrangian form could then be denoted as:
> > >
> > > $$\min_{f: X\to [K]} ~\Phi(f\_{k}):=\mathbb{E}\_{(X,  \widetilde{Y})\sim \widetilde{\mathcal{D}}}[\ell(f\_{k}(X), \widetilde{Y})] + \frac{\mu\_{k}}{2} \cdot \sum\_{i\in[N]}{\text{Dist}\_{i}}^2+ \sum\_{i\in[N]}\lambda_i \cdot  \text{Dist}\_{i}$$
> > >
> > > with $\lambda_i$ updating as $\lambda\_{i}\leftarrow \lambda\_{i} + \mu\_{k}\cdot \text{Dist}\_{i}$, and $\text{Dist}\_{i}$ takes the input classifier $f\_{k}$ given by the solution to Eqn. (R1) at the $k$-th iteration.
> > >
> > > In such an augmented Lagrangian method, the variable $\lambda$ could be viewed as an estimate of the Lagrange multiplier, and the accuracy of this estimate keeps improving as iteration progresses. We adopt the augmented format instead of the penalty method for the reason that: it is not necessary to take $\mu \rightarrow \infty$ to solve Eqn. (2). And the use of the Lagrange multiplier term allows us to choose smaller $\mu_k$, hence avoiding ill-conditioning.
> > >
> > > Since we are not requiring strict fair performance among sub-populations, setting a constant $\lambda$ tends to be a simple implementation to control the performance gap. Our ablation study (Table 3) on constant $\lambda$ further demonstrates the effectiveness of constant $\lambda$s in FR.
> > >
> > > Regarding your concern [“Applying dual ascent is likely to result in a larger $\lambda_i$ on the worst sub-population”, ]
> > >
> > > This is because worst sub-population $i$ has the lowest accuracy, which results in a larger performance gap w.r.t. the overall performance, mathematically,
> > >
> > > $$\text{Dist}\_{i}=|\mathbb{P}(f(X)=\widetilde{y}|G=i) - \mathbb{P}(f(X)=\widetilde{y})|.$$
> > >
> > > In the risk optimization view, due to the appearance of Fairness Regularizer, the model will have overall higher expected priority (larger weights) for samples in this sub-population $i$, compared with other populations, since population $i$ has the largest performance gap.
> > >
> > > ```Question 4:```
> > >
> > > [Eq. 3: given that the constraint that is actually used in practice is different from the definition of $\text{Dist}_i$ given in Section 4.1., I'd suggest giving a different name to each of these constraints.]
> > >
> > > ```Response:```
> > >
> > > Thanks for the suggestion, we will change the name of $\text{Dist}_i$ to avoid any confusion!

---

> > > > ### Comment · Reviewer_Ed4g · 2023-11-23
> > > >
> > > > Thank you for the detailed response.
> > > >
> > > > I think the paper's motivation is strong, but the experiments section should be significantly improved to support the main claim.
> > > >
> > > > In my opinion it should position itself as an empirical paper that tests/shows that fairness-aware learning improves learning on noisily labelled long tailed data. Any focus given to the fairness regularizer distracts from the main point, as the FR is not novel. I also agree with W1 of Reviewer yYZa.

---

### Official Review · Reviewer_MjTZ · 2023-10-31

**Soundness:** 3 good
**Presentation:** 2 fair
**Contribution:** 2 fair
**Rating:** 5
**Confidence:** 4

**Summary:**

The paper explores the problem of learning from noisy labeled and long-tailed data. The paper starts by observing that existing methods to improve learning from noisy labeled or long-tailed data create accuracy disparities across different sub-populations in the dataset. The paper proposes a method called Fairness Regularizer (FR) that incentivizes the learned classifier to have uniform performance across subpopulations, aiming to mitigate accuracy disparities and improve learning. Experiments indicate that the proposed method improves the model performance in underrepresented populations in the dataset while increasing overall model performance.

**Strengths:**

•	The idea of connecting fairness with robust learning is exciting and could lead to interesting results.

•	The discovery in Figure 2 of the disparate impact in underrepresented groups when using methods to learn from data with label noise or long-tail is interesting and should be communicated to the community.

•	The conclusion that methods that incentivize statistical parity led to better model performance under noisy labels and long-tail data is very nice. The results contradict the folklore that fairness improvement methods *always* decrease the model performance.

•	The results demonstrated in the paper had strong statistical guarantees, as shown in Table 2.

**Weaknesses:**

•	The proposed method, Fairness Regularizer, is similar to the reductions approach for statistical parity [1]. But I didn’t find a citation for the paper.

•	There are multiple existing in-processing methods to ensure fairness (statistical parity) across groups in the population check [2] Table 1 for multiple references. The authors should, ideally, compare their results with other fairness improvement methods. At least, the paper should mention other fairness improvement methods and discuss the difference between the proposed and existing approaches.

•	I believe the points in the lower-left corner of Figure 2 and 6 are the underrepresented groups. However, Figure 6 and Figure 2 could provide a more explicit message to the reader by adding one more axis of information (e.g., the intensity of the color of the points) with the group representation.

**Questions:**

•	Could the authors please highlight the main differences between FR and the reductions approach [1]?

•	Why is the proposed method preferred compared to other fairness (statistical parity) improvement methods?



[1] Agarwal et al. A Reductions Approach for Fair Classification. 2018.

[2] Lowy et al.  A Stochastic Optimization Framework for Fair Risk Minimization. 2022

---

> ### Author Response · Authors · 2023-11-21
> **Response to Reviewer MjTZ (Part 1)**
>
> Thanks for your time for reviewing our work! Please refer to our following responses regarding your insightful suggestions!
>
> ```Weaknesses 1:```
>
> [The proposed method, Fairness Regularizer, is similar to the reductions approach for statistical parity [1]. But I didn’t find a citation for the paper.]
>
> ```Response:```
>
> Thanks for the great suggestions. In our revision, we will explicitly distinguish between our proposed Fairness Regularizer (FR) and the reductions approach for statistical parity, as outlined in Agarwal et al. (2018).
>
> To briefly clarify, while both methods share a common goal of promoting fairness, they differ significantly in their implementation and outcomes. Traditional reduction approaches, including the one mentioned in your reference, predominantly focus on achieving statistical equality across different groups. This is often achieved by modifying the training process, specifically by addressing the Lagrangian:
>
> $$\mathbb{E}\_{(X,  \widetilde{Y})\sim \widetilde{\mathcal{D}}}[\ell(f(X), \widetilde{Y})] +\sum\_{i=1}\^{N}\lambda\_{i} (\text{Dist}\_{i}-\delta),$$
>
> which aims to offer an alternative solution to the original optimization problem presented in Equation (2):
>
> $$\min\_{f:  X\rightarrow [K]} \mathbf{E}\_{(X, \widetilde{Y})\sim \widetilde{\mathcal{D}}} [\ell(f(X), \widetilde{Y})], \qquad s.t. \qquad \text{Dist}\_{i}\leq \delta, \forall i \in [N], \text{[Eqn. (2)]} $$
>
> In contrast, our FR approach, while superficially similar, diverges in its core methodology. Specifically, it entails optimizing the Lagrangian in Equation (2) with uniform $\lambda_i=\lambda$ and $\delta=0$. This distinction is crucial as it results in different optimal solutions. Our approach, aimed at enhancing overall performance on clean and balanced test data, strategically avoids overemphasis on the worst-performing sub-populations. Unlike reduction approaches that strive for strict fairness across all sub-populations, our method prioritizes a more balanced improvement across the board.
>
> We will ensure to elaborate on this point more clearly in our revised manuscript, including appropriate citations to Agarwal et al. (2018) to acknowledge their foundational work in this area. Thank you once again for your valuable feedback, and we look forward to incorporating these clarifications in our revision.
>
> [1] Agarwal et al. A Reductions Approach for Fair Classification. 2018.
>
> ```Weaknesses 2:```
>
> [There are multiple existing in-processing methods to ensure fairness (statistical parity) across groups in the population check [2] Table 1 for multiple references. The authors should, ideally, compare their results with other fairness improvement methods. At least, the paper should mention other fairness improvement methods and discuss the difference between the proposed and existing approaches.]
>
> ```Response:```
>
> Thank you for your valuable feedback. We appreciate your insights and agree that a comparison with existing fairness improvement methods would enrich our paper. Please allow us to clarify our approach in light of your comments.
>
> **Clarification on Our Focus:** Our primary objective is not solely to achieve fairness, but to demonstrate how fairness measures can enhance learning with noisy long-tail data, thereby improving the model's overall performance while reducing disparities across groups. Our method draws inspiration from popular in-processing fairness solutions, particularly those involving fairness definitions and constraints. However, our aim is distinct: we focus on enhancing overall performance rather than solely achieving the highest level of fairness.
>
> **Discussion on Methodological Differences:** We recognize the importance of situating our work within the broader context of fairness in machine learning. To this end, we will include a detailed discussion of various fairness improvement methodologies in our revised manuscript. Here is a brief overview of the additional related works we intend to incorporate:
>
> [More in Part 2 responses~]

---

> ### Author Response · Authors · 2023-11-21
> **Response to Reviewer MjTZ (Part 2)**
>
> We continue our response as below:
>
> Here is a brief overview of the additional related works we intend to incorporate:
>
> * Pre-Processing Techniques: These methods, like reweighing [1] and optimized pre-processing [2], aim to reduce bias in training data by adjusting training instance weights and modifying feature values for balanced representation.
>
> * Post-Processing Techniques: Techniques such as those proposed by Hardt et al. (2016) [5] adjust model predictions after training to achieve fairness objectives, often focusing on equal error rates across groups.
>
> * In-Processing Techniques: Central to our research, these strategies involve modifying the learning algorithm itself. Examples include fairness constraints within the optimization objective [3] and adversarial debiasing [4].
>
> * Our proposed Fairness Regularizer aligns with in-processing methods but is unique in its pursuit of optimal performance on clean and balanced test data. It aims for a balance in performance across sub-populations, diverging from reductionist approaches like Agarwal et al. (2018) [6] by prioritizing overall efficacy alongside fairness.
> We hope this additional context addresses your concerns. We are also open to suggestions for any important related works in fairness machine learning that we may have overlooked.
>
>
>
> References
>
> [1] Kamiran, F., & Calders, T. (2012). Data preprocessing techniques for classification without discrimination. Knowledge and Information Systems, 33(1), 1-33.
>
> [2] Calmon, F., Wei, D., Ramamurthy, K. N., Varshney, K. R., & Rosa, A. (2017). Optimized pre-processing for discrimination prevention. Advances in Neural Information Processing Systems, 30.
>
> [3] Zafar, M. B., Valera, I., Gomez Rodriguez, M., & Gummadi, K. P. (2017). Fairness constraints: Mechanisms for fair classification. In Proceedings of the 20th International Conference on Artificial Intelligence and Statistics (AISTATS).
>
> [4] Zhang, B., Lemoine, B., & Mitchell, M. (2018). Mitigating unwanted biases with adversarial learning. In Proceedings of the 2018 AAAI/ACM Conference on AI, Ethics, and Society.
>
> [5] Hardt, M., Price, E., & Srebro, N. (2016). Equality of opportunity in supervised learning. Advances in Neural Information Processing Systems, 29.
>
> [6] Agarwal, A., Beygelzimer, A., Dudík, M., Langford, J., & Wallach, H. (2018). A reductions approach to fair classification. In International Conference on Machine Learning (ICML).
>
>
> ```Weaknesses 3:```
>
> [I believe the points in the lower-left corner of Figure 2 and 6 are the underrepresented groups. However, Figure 6 and Figure 2 could provide a more explicit message to the reader by adding one more axis of information (e.g., the intensity of the color of the points) with the group representation.]
>
> ```Response:```
>
> Thanks for this insightful suggestion. We will revise the figure by adding one more axis of information with the group representation
>
> ```Question 1:```
>
> [Could the authors please highlight the main differences between FR and the reductions approach [1]?]
>
> ```Response:```
>
> Please refer to our response to weakness 1.
>
> ```Question 2:```
>
> [Why is the proposed method preferred compared to other fairness (statistical parity) improvement methods?]
>
> [1] Agarwal et al. A Reductions Approach for Fair Classification. 2018.
> [2] Lowy et al. A Stochastic Optimization Framework for Fair Risk Minimization. 2022
>
> ```Response:```
>
> Thank you for raising this insightful question. While the solutions proposed by Agarwal et al. (2018) and Lowy et al. (2022) are indeed efficient in targeting fair risk minimization, our approach diverges in its primary objectives. Our research is not focused on achieving fairness per se, but on demonstrating how fairness measures can be instrumental in learning from noisy, long-tail data. The goal is to enhance the model's overall performance, which incidentally also helps in reducing disparities across groups.
>
> Furthermore, our methodology draws inspiration from one of the most recognized in-processing fairness solutions—implementing constraints based on fairness definitions. However, unlike these solutions, our emphasis is not solely on attaining the highest level of fairness but on leveraging these fairness measures as a means to improve overall model performance. This subtle but significant shift in focus distinguishes our method from the ones mentioned and aligns it more closely with our research objectives.
>
> We appreciate your interest in our approach and hope this clarification helps in understanding the distinct positioning of our method in the landscape of fairness improvement strategies.

---

> > ### Comment · Reviewer_MjTZ · 2023-11-22
> >
> > Thank you, authors, for your response.
> >
> > Given its claim, this paper could significantly impact the community. However, it still needs to be substantially improved to support its claims.
> > For this reason, I will maintain the scores I provided.

---

### Official Review · Reviewer_ziso · 2023-11-02

**Soundness:** 2 fair
**Presentation:** 3 good
**Contribution:** 3 good
**Rating:** 6
**Confidence:** 2

**Summary:**

The paper considers a challenging learning problem under the existence of long-tailed and noisily labeled data. The paper makes efforts to understand and mitigate the heterogeneous effects of label noise for the imbalaenced data. The paper empirically examines the role of each sub-population and its influence on the test data. It proposes a fairness regularizer to encourage the learned classifier to achieve fair performance across different sub populations.

**Strengths:**

The paper is generally well written. The structure is clear and the motivation is strong. The Fairness Regularizer (FR) is easy to implement. The numerical results are relatively complete. Some theoretical insights are also included in the appendices.

**Weaknesses:**

The paper overally has no big weakness.

Minor suggestions:

(i) How to tune $\lambda_i$'s? Is there any recommendation or theory to support this?
(ii) Some theory presented in the appendix can be moved into main content.
(iii) Can author make some comments on distribution shift problem? If some sub-population in the test data does not appear in training data, how to guarantee its performance?

**Questions:**

See weakness points.

**Details Of Ethics Concerns:**

No ethics concerns.

---

> ### Author Response · Authors · 2023-11-21
> **Response to Reviewer ziso**
>
> Thanks for your time for reviewing our work! Please refer to our following responses regarding your insightful suggestions!
>
> ```Weaknesses 1:```
>
> [How to tune lambda ? Is there any recommendation or theory to support this?]
>
> ```Response:```
>
> Thank you for your insightful query regarding the tuning of $\lambda$. We appreciate the opportunity to clarify this aspect of our study.
>
> In our research, we did not focus on providing a theoretical basis for $\lambda$ tuning, primarily because our approach does not necessitate strict fairness across sub-populations. In our experimental framework, we opted for a straightforward approach by selecting constant $\lambda$ values. This decision was driven by practical observations, where we found that using a constant $\lambda$ generally yields a simple yet effective means to manage the performance disparity.
>
> For a more comprehensive understanding, we included an ablation study of $\lambda$ in Table 3, utilizing the Clothing 1M dataset. In this study, we explored various $\lambda$ values, specifically chosen from the set {0.0,0.1,0.2,0.4,0.6,0.8,1.0,2.0}. It’s important to note that a $\lambda$ value of 0.0 corresponds to the baseline methods' training without the application of our Fairness Regularizer (FR).
>
> Our findings suggest that the default setting for FR ($\lambda=1.0$) consistently achieves competitive performance compared to other $\lambda$ values. This observation supports our recommendation to adopt $\lambda=1.0$ as a practical default in similar applications. Additionally, our results indicate that $\lambda$ values close to 1.0 generally outperform those nearer to 0.0. This trend not only underscores the effectiveness of our proposed fairness regularizer but also suggests a relative insensitivity of the model to hyper-parameter variations near the chosen $\lambda$ value.
>
> ```Suggestion 1:```
>
> [Some theory presented in the appendix can be moved into main content]
>
> ```Response:```
>
> Thanks for the suggestion. In our revision, we will consider summarizing main insights of our theoretical results in the appendix into the main content!
>
> ```Question 1:```
>
> [Can the author make some comments on the distribution shift problem? If some sub-population in the test data does not appear in training data, how to guarantee its performance?]
>
> ```Response:```
>
> This is a great question! When certain subpopulations $G_i$ does not appear in the training data, to improve the performance of $G_i$ at test time, it makes sense to assume that we have a held-out validation set for model selection/calibration. In that case, we could perform some post-hoc treatments on model weights scaling at test time, such weights could be inspired by referring to a validation set. Meanwhile, we tend to not intervene in the training process (except for FR) for the sake of fitting well on the existing sub-populations.

---

### Official Review · Reviewer_yYZa · 2023-11-03

**Soundness:** 2 fair
**Presentation:** 3 good
**Contribution:** 2 fair
**Rating:** 3
**Confidence:** 3

**Summary:**

This paper is trying to address an interesting problem, that is, how to eliminate long-tailed effect and noisy labels simultaneously. Previous approaches have often tackled these issues independently and have not considered the combined effects of both. This paper empirically demonstrates that such isolated solutions are ineffective when dealing with long-tailed datasets that also have label noise. Furthermore, existing methods do not consistently improve accuracy across all sub-populations, leading to uneven performance improvements. In response to these observations, the paper introduces the Fairness Regularizer (FR), which aims to reduce performance disparities between sub-populations. FR is shown to enhance the performance of tail sub-populations and overall learning in the presence of label noise. Experiment results support the effectiveness of this approach, especially when combined with existing robust or class-balanced methods.

**Strengths:**

S1: This paper introduces a novel perspective on addressing both long-tailed problem and noisy label problem, which is rarely discussed in previous studies.

S2: This paper provides some empirical evidence, which helps to better interpret motivation.

S3: This paper is well-organized and clearly written.

**Weaknesses:**

W1: The logic of this paper is rather confusing; the author discusses the issues of noisy data and long tail problem together, but does not build a good bridge of them. In addition, the significance and necessity of the empirical evidence is not clear. Because it is also necessary to deal with the long tail problem on clean data, instead of only need to deal with the long tail problem on noisy data. The authors need to emphasize the differences and connections between dealing with long tail problems on noisy data and dealing with long tail problems under clean data (e.g., problem formalization or techniques).

W2: Why observation 3.1 holds？Specifically, if prediction model trained on noisy data has larger variance than the prediction model trained on clean data, the same pattern will be also observed. Therefore, observation 3.1 depends on the original variance of the prediction model. In addition, if we did not remove any sub-population and draw an Acc plot for clean data and noisy data, can we observe the similar pattern in Figure 4?  Similar arguments are also holds for observation 3.2.

W3: The problem is handled by simply adding the distance of each class from the "center" to the original loss function, which is simple and lacks some novelty. For example, is it possible to calibrate clean data with noisy data or head with tail data to address this problem?

W4: How can we choose $\lambda_{i}$ for each sample in practice when it is not set to a constant?

W5: This paper is missing some noisy label and long-tail baselines. There are many works trying to denoise, like [1-4], and many long-tail baselines like [5-6]. I didn't intend for the authors to compare all noisy label and long-tail baselines, but it is necessary to consider some of them.

[1] Patrini, Giorgio, et al. "Making deep neural networks robust to label noise: A loss correction approach." in CVPR 2017

[2] Han, Bo, et al. "Co-teaching: Robust training of deep neural networks with extremely noisy labels." in NeurIPS 2018.

[3] Xia, Xiaobo, et al. “Are anchor points really indispensable in label-noise learning?.” in NeurIPS 2019.

[4] Yong, Lin., et al. "A Holistic View of Label Noise Transition Matrix in Deep Learning and Beyond." in ICLR 2022.

[5] Chen, Wei, et al. "Crest: A class-rebalancing self-training framework for imbalanced semi-supervised learning. in CVPR 2021.

[6] Zhong, Zhisheng, et al. "Improving Calibration for Long-Tailed Recognition" in CVPR 2021.

**Questions:**

Please refer to weakness part for the questions.

---

> ### Author Response · Authors · 2023-11-21
> **Response to Reviewer yYZa (Part 1)**
>
> Thanks Reviewer yYZa for the efforts and time in reviewing our paper! Please see our response below regarding your concerns and mentioned weaknesses.
>
> ```Weaknesses 1:```
>
> [The authors need to emphasize the differences and connections between dealing with long tail problems on noisy data and dealing with long tail problems under clean data (e.g., problem formalization or techniques).]
>
>
> ```Response:```
>
> **Regarding Problem Formalization:**
> We appreciate the reviewer's point on differentiating between long-tail problems in clean versus noisy data contexts. In our perspective, addressing long-tail issues in clean data can be considered a subset of the broader challenge posed by noisy data. This relationship becomes apparent when considering the transition from clean to noisy labels, represented as $T_{i, j}(X)=\mathbb{P}(\widetilde{Y}=j | Y=i, X)$. In this framework, the scenario with clean data emerges when $T(X)$ simplifies to an identity matrix.
>
> **Concerning Techniques**
> For illustrative purposes, we refer to the classical setting with class-dependent transition probabilities ($T_{i, j}(X)=T_{i, j}$, irrespective of $X$). Here, the learning or optimization tasks differ significantly between clean and noisy label scenarios. In the clean label context, the objective is to minimize:
>
> $\mathbf{E}_{(X, Y)\sim \mathcal{D}} \left[\ell(f(X), Y)\right],$
>
> which is equivalent to
>
> $\sum_{i} \mathbf{E}_{X} ~ \mathbb{P}(Y=i)  \left[\ell(f(X), Y=i)\right].$
>
> In contrast, with noisy labels, the objective function shifts to minimizing:
>
> $$\mathbf{E}_{(X, \widetilde{Y})\sim \widetilde{\mathcal{D}}} ~\left[\ell(f(X), \widetilde{Y})\right].$$
>
> The objective under the label noise setting is equivalent to:
>
> $$\sum_j  \mathbf{E}_{X} ~\mathbb{P}(\widetilde{Y}=j) \left[\ell(f(X), \widetilde{Y}=j)\right].$$
>
> Leveraging the transition matrix, we could then derive:
>
> $$\sum_j \sum_i  \mathbf{E}_{X} ~\mathbb{P}(\widetilde{Y}=j|Y=i) \mathbb{P}(Y=i)  \left[\ell(f(X), \widetilde{Y}=j)\right],$$
>
> It is worthy noting that the impact of class-imbalanced distribution is indicated in the formulation as well (varied $\mathbb{P}(Y=i)$). As we can see the differences, with the presence of label noise, the technique differences lie in how to further mitigate/correct the impact brought by $T_{i, j}=\mathbb{P}(\widetilde{Y}=j|Y=i)$.
>
> ```Weakness 2:```
>
>  [Why observation 3.1 holds？Specifically, if prediction model trained on noisy data has larger variance than the prediction model trained on clean data, the same pattern will be also observed. Therefore, observation 3.1 depends on the original variance of the prediction model. In addition, if we did not remove any sub-population and draw an Acc plot for clean data and noisy data, can we observe the similar pattern in Figure 4? Similar arguments are also held for observation 3.2.]
>
> ```Response:```
>
> Thank you for your insightful comments and queries regarding Observation 3.1 in our manuscript. We appreciate the opportunity to clarify these points and ensure the accuracy of our findings.
> Firstly, we acknowledge your perspective on the potential variance of prediction models trained on noisy versus clean data. However, the focus of Observation 3.1 and Figure 4 in our study is not on the variance of model predictions but on the performance disparity, specifically the test accuracy drops at the sub-population level. Our analysis is centered on comparing the following metrics：
>
> $\text{Acc}\_{\text{p}}(\mathcal{A},S, i, j)$  v.s.  $\text{Acc}\_{\text{p}}(\mathcal{A}, \widetilde{S}, i, j)$.
>
> In Figure 4, the broader distribution of the box plot for  $\text{Acc}\_{\text{p}}(\mathcal{A}, \widetilde{S}, i, j)$  (pertaining to the noisy-label scenario) as compared to $\text{Acc}\_{\text{p}}(\mathcal{A},S, i, j),$ indicates a more pronounced influence of noisy data, thereby reinforcing Observation 3.1.
>
> Regarding your question about the absence of sub-population removal and its effect on the accuracy plot for clean versus noisy data, we have addressed this in Figure 9 (located in the Appendix, lower right corner). It appears there may have been a misunderstanding regarding the definition of our influence measure depicted in Figure 4, which focuses on the accuracy drop. To elucidate, when calculating  $\text{Acc}\_{\text{p}}(\mathcal{A}, \widetilde{S}, i, j)$ for each $i, j$, we assess the following difference:
>
> $$\mathbb{P}\_{f \leftarrow \mathcal{A} (\widetilde{S}), (X’, Y’, G=j)} (f(X’)=Y’) - \mathbb{P}\_{f \leftarrow \mathcal{A}(\widetilde{S}^{\backslash i}), (X’, Y’, G=j)} (f(X’)=Y’)$$
>
> Here, the upper term represents the scenario where no sub-populations are removed in training. Following your query, if no subpopulation is excluded, the lower term becomes identical to the first, resulting in a net influence of zero.

---

> ### Author Response · Authors · 2023-11-21
> **Response to Reviewer yYZa (Part 2)**
>
> ```Weakness 3:```
>
> [The problem is handled by simply adding the distance of each class from the "center" to the original loss function, which is simple and lacks some novelty. For example, is it possible to calibrate clean data with noisy data or head with tail data to address this problem?]
>
> ```Response:```
>
> Thank you for your insightful feedback. We appreciate your suggestion to calibrate clean data with noisy data or head with tail data. Indeed, this approach offers an intriguing perspective. However, our decision to not pursue this direction was driven by several pragmatic considerations.
>
> Firstly, implementing such a solution would necessitate additional information, such as clean labels for training samples or a separate, clean validation set. Our study operates under the realistic assumption that these resources are not readily available. Estimating the necessary statistics without these could be particularly challenging in the context of long-tailed data, potentially leading to errors that might compromise robust learning.
>
> Furthermore, our empirical observations highlighted the importance of controlling the performance disparity among different populations. Introducing a fairness regularizer aligned more closely with our research narrative and objectives, offering a more feasible solution under the constraints of our study.
>
> Regarding the simplicity of our method, we believe that simplicity does not detract from novelty, particularly when it effectively addresses the complexities of long-tail distribution and label noise. The "distance" we refer to in our approach is not a traditional $L_{1}, L_{2}$ distance, but rather a measure of the performance gap across sub-populations. To our knowledge, our work is unique in employing fairness regularization to control this gap under such challenging conditions.
>
> ```Weakness 4:```
>
> [How can we choose lambda for each sample in practice when it is not set to a constant?]
>
> ```Response:```
> Thank you for your insightful question regarding the selection of lambda for each sample in our proposed method. Your query touches on a fundamental aspect of our approach, which we appreciate.
> In addressing the optimization task presented in Equation (2):
>
>
> $$\min_{f:  X\rightarrow [K]} \mathbf{E}_{(X, \widetilde{Y})\sim \widetilde{\mathcal{D}}} [\ell(f(X), \widetilde{Y})],  \qquad s.t.  \qquad\text{Dist}_i\leq \delta, \forall i \in [N], \qquad \text{[Eqn. (2)]} $$
>
> We could find an empirical solution utilizing the augmented Lagrangian form. This approach transforms the constrained problem into a series of unconstrained minimization problems, as detailed in Equation (R1):
>
> $$\min_{f:  X\rightarrow [K]}  \Phi(f_k):= \mathbf{E}_{(X, \widetilde{Y})\sim \widetilde{\mathcal{D}}}  [\ell(f\_{k} (X), \widetilde{Y})]+  \mu\_{k} \cdot \sum\_{i\in[N]}  g( \text{Dist}\_{i}-\delta), \text{[Eqn. (R1)]}$$
>
> where $g(x):=\max(0, x)^2$. The iterative process involves updating the solution from each iteration to serve as the initial guess for the subsequent one. This method ensures that the solutions converge asymptotically to the optimization task defined in Equation (2).
>
> In the augmented Lagrangian form, with $\delta=0$, the formulation becomes:
>
> $$\min_{f: X\to [K]} ~\Phi(f_k):=\mathbb{E}_{(X,  \widetilde{Y})\sim \widetilde{\mathcal{D}}}[\ell(f\_{k}(X), \widetilde{Y})] + \frac{\mu\_{k}}{2} \cdot \sum\_{i\in[N]}{\text{Dist}\_{i}}^2+ \sum\_{i\in[N]} \lambda\_{i} \cdot  \text{Dist}\_{i}$$
>
> with $\lambda_i$ updating as $\lambda\_{i}\leftarrow \lambda\_{i} + \mu\_{k}\cdot \text{Dist}\_{i}$, and $\text{Dist}\_{i}$ takes the input classifier $f_k$ given by the solution to Eqn. (R1) at the $k$-th iteration. In such an augmented Lagrangian method, the variable $\lambda$ could be viewed as an estimate of the Lagrange multiplier, and the accuracy of this estimate keeps improving as iteration progresses. We adopt the augmented format instead of the penalty method for the reason that: it is not necessary to take $\mu \rightarrow \infty$ to solve Eqn. (2). And the use of the Lagrange multiplier term allows us to choose smaller $\mu_k$, hence avoiding ill-conditioning.
>
> Regarding the adoption of a constant $\lambda$, our approach stems from the lack of a strict requirement for equal performance across sub-populations. A constant $\lambda$ offers a simpler implementation to manage performance disparities. Our ablation study (referenced in Table 3) supports this choice, demonstrating the efficacy of constant $\lambda$ values in Fairness Regularization (FR).

---

> > ### Author Response · Authors · 2023-11-21
> > **Response to Reviewer yYZa (Part 3)**
> >
> > ```Weakness 5:```
> >
> > This paper is missing some noisy labels and long-tail baselines.
> >
> > ```Response:```
> >
> > Thank you for your insightful feedback. We acknowledge the importance of considering noisy labels and long-tail baselines in our study. Prior to submitting our paper, we conducted extensive experiments with various robust loss designs, such as backward loss correction [1], forward loss correction [1], and GCE loss [2], alongside class-imbalanced learning solutions like class-balanced loss [3] and LDAM loss [4]. However, our findings revealed that these methods significantly underperformed compared to the CE loss, particularly when confronted with the combined challenges of long-tail distribution and label noise.
> > We deeply appreciate your suggestions and take them seriously. In light of your feedback, we are actively exploring additional methods, including those you've mentioned. We believe this ongoing research will further enrich our understanding and enhance the robustness of our approach. We are grateful for the opportunity to improve our work and look forward to incorporating these insights into our research.
> >
> > **References:**
> >
> > [1] Patrini, Giorgio, et al. "Making deep neural networks robust to label noise: A loss correction approach." CVPR 2017
> >
> > [2] Zhang, Zhilu, and Mert Sabuncu. "Generalized cross entropy loss for training deep neural networks with noisy labels." NeurIPS 2018.
> >
> > [3] Cui, Yin, et al. "Class-balanced loss based on effective number of samples." CVPR, 2019.
> >
> > [4] Cao, Kaidi, et al. "Learning imbalanced datasets with label-distribution-aware margin loss." NeurIPS 2019.

---

### Meta-Review · Area_Chair_77oT · 2023-12-11

**Metareview:**

This paper uses tools from fair machine learning literature to tackle issues due to long-tailed data and noisy labels simultaneously. While reviewers appreciate the simplicity of the approach, they also brought up some issues. Reviewers found the relationship between noisy data issues and the long-tail problem confusing. The paper could clarify and differentiate the approaches for noisy data and clean data. The empirical results are not particularly strong.

The reviewers also made some concrete suggestions for improvements. For example, some theoretical results can be from the appendices to the main body of the paper. There should be a further discussion of how the proposed fairness regularize is related to prior work on fairness, especially the reduction approach.

In summary, while the paper introduces a new approach to a less explored problem, it still requires improvements in clarity, depth, and stronger empirical evidence.

**Justification For Why Not Higher Score:**

The AC also read the paper and agreed with the assessment of the reviewers.

**Justification For Why Not Lower Score:**

N/A

---

### Decision · Program_Chairs · 2024-01-16

Reject